

# Characterization of Transport Regimes and the Polar Dome during Arctic Spring and Summer using in-situ Aircraft Measurements

Heiko Bozem[1], Peter Hoor[1], Daniel Kunkel[1], Franziska Köllner[1,2], Johannes Schneider[2], Andreas Herber[3], Hannes Schulz[3], W. Richard Leaitch[4], Amir A. Aliabadi[5], Megan D. Willis[6,*], Julia Burkart[6,**], and Jonathan P. D. Abbatt[6]

[1]Johannes Gutenberg University of Mainz, Institute for Atmospheric Physics, Mainz, Germany
[2]Particle Chemistry Department, Max Planck Institute for Chemistry, Mainz, Germany
[3]Alfred Wegener Institute Helmholtz Centre for Polar and Marine Research, Bremerhaven, Germany
[4]Environment and Climate Change Canada, Toronto, Canada
[5]School of Engineering, University of Guelph, Guelph, ON, Canada
[6]Department of Chemistry, University of Toronto, Toronto, Canada
[*]now at: Chemical Sciences Division, Lawrence Berkeley National Laboratory, Berkeley, California, USA
[**]now at: University of Vienna, Aerosol Physics & Environmental Physic, Vienna, Austria

**Correspondence:** Heiko Bozem (bozemh@uni-mainz.de)

**Abstract.** The springtime composition of the Arctic lower troposphere is to a large extent controlled by transport of mid-latitude air masses into the Arctic, whereas during the summer precipitation and natural sources play the most important role. Within the Arctic region, there exists a transport barrier, known as the polar dome, which results from sloping isentropes. The polar dome, which varies in space and time, exhibits a strong influence on the transport of air masses from mid-latitudes, enhancing

it during winter and inhibiting it during summer. Furthermore, a definition for the location of the polar dome boundary itself is quite sparse in the literature.

We analyzed aircraft based trace gas measurements in the Arctic during two NETCARE airborne field campaigns (July 2014 and April 2015) with the Polar 6 aircraft of Alfred Wegener Institute Helmholtz Center for Polar and Marine Research (AWI), Bremerhaven, Germany, covering an area from Spitsbergen to Alaska ($134\,°W$ to $17\,°W$ and $68\,°N$ to $83\,°N$). For the spring

(April 2015) and summer (July 2014) season we analyzed transport regimes of mid-latitude air masses travelling to the high Arctic based on CO and $CO_2$ measurements as well as kinematic 10-day back trajectories. The dynamical isolation of the high Arctic lower troposphere caused by the transport barrier leads to gradients of chemical tracers reflecting different local chemical life times and sources and sinks. Particularly gradients of CO and $CO_2$ allowed for a trace gas based definition of the polar dome boundary for the two measurement periods with pronounced seasonal differences. For both campaigns a transition

zone rather than a sharp boundary was derived. For July 2014 the polar dome boundary was determined to be $73.5\,°N$ latitude and $299 - 303.5\,K$ potential temperature, respectively. During April 2015 the polar dome boundary was on average located at $66 - 68.5\,°N$ and $283.5 - 287.5\,K$. Tracer-tracer scatter plots and probability density functions confirm different air mass properties inside and outside of the polar dome for the July 2014 and April 2015 data set. Using the tracer derived polar dome boundaries the analysis of aerosol data indicates secondary aerosol formation events in the clean summertime polar dome.





Synoptic-scale weather systems frequently disturb this transport barrier and foster exchange between air masses from mid-latitudes and polar regions. During the second phase of the NETCARE 2014 measurements a pronounced low pressure system south of Resolute Bay brought inflow from southern latitudes that pushed the polar dome northward and significantly affected trace gas mixing ratios in the measurement region. Mean CO mixing ratios increased from $77.9 \pm 2.5\,\mathrm{ppbv}$ to $84.9 \pm 4.7\,\mathrm{ppbv}$

from the first period to the second period. At the same time $CO_2$ mixing ratios significantly dropped from $398.16 \pm 1.01\,\mathrm{ppmv}$ to $393.81 \pm 2.25\,\mathrm{ppmv}$.

We further analysed processes controlling the recent transport history of air masses within and outside the polar dome. Air masses within the spring time polar dome mainly experienced diabatic cooling while travelling over cold surfaces. In contrast air masses in the summertime polar dome were diabatically heated due to insolation. During both seasons air masses outside

the polar dome slowly descended into the Arctic lower troposphere from above caused by radiative cooling. The ascent to the middle and upper troposphere mainly took place outside the Arctic, followed by a northward motion. Our results demonstrate the successful application of a tracer based diagnostic to determine the location of the polar dome boundary.

## 1   Introduction

In recent decades the Arctic has undergone dramatic changes affecting sea ice, snow, permafrost, surface temperature, land,

snow and atmospheric circulation (IPCC, 2013). Rising temperatures, twice as fast as in the rest of the world, lead to a significant retreat of Arctic sea ice (Stroeve et al., 2012; Jeffries et al., 2013). In addition to the reduced extent the thickness of sea ice is continuously decreasing (Lindsay and Schweiger, 2015). The continuing retreat of Arctic sea ice will increase the accessibility of the Arctic thus leading to a potential increase of emissions from local sources of pollutants like shipping (Eckhardt et al., 2013; Corbett et al., 2010; Melia et al., 2016) and oil and gas extraction (Peters et al., 2011). Already to date

atmospheric pollutants such as aerosol particles and tropospheric ozone contribute to Arctic warming (Shindell and Faluvegi, 2009; AMAP, 2015). It was shown by earlier studies that mid-latitude emissions in the Northern Hemisphere are the main source region for atmospheric pollutants in the Arctic (Barrie, 1986; Koch and Hansen, 2005; Stohl, 2006; Sharma et al., 2013; Arnold et al., 2016). Several studies either based on in-situ measurements or modelling reported enhanced pollution throughout the Arctic troposphere that is dominated by northern Eurasian sources in the lower troposphere and mid-latitude

North America and Asia above (Sharma et al., 2006; Shindell et al., 2008; Fisher et al., 2010; Hirdman et al., 2010; Hecobian et al., 2011; Brock et al., 2011; Schmale et al., 2011; Sodemann et al., 2011; Stohl et al., 2013; Law et al., 2014; Monks et al., 2015). Roiger et al. (2011) even found Asian pollution in the lowermost stratosphere. However, local emissions in specific regions within the Arctic are already important (Stohl et al., 2013) and might gain influence in the near future (Corbett et al., 2010; Peters et al., 2011). Compared to other regions of the Northern Hemisphere, the faster pace of the rising surface and

lower tropospheric temperatures in the Arctic is commonly known as Arctic amplification (Holland and Bitz, 2003; Screen and Simmonds, 2010). The interplay between different processes fosters feedback mechanisms that further amplify changes in the environment. The decrease in sea ice reduces the surface albedo and increases latent and sensible heat fluxes into the atmosphere which in turn results in warmer surface temperatures relative to mid-latitudes (Robock, 1983; Hall, 2004; Winton,





2006). Furthermore, Pithan and Mauritsen (2014) reported that also temperature feedbacks play an important role for Arctic amplification. Arctic Amplification could further cause important changes in the mid-latitude circulation (Cohen et al., 2014; Pithan et al., 2018). Zonal winds might weaken and the Rossby wave amplitude is supposed to increase especially during the fall and winter months (Francis and Vavrus, 2012; Francis et al., 2017). Unusual warm sea surface temperatures and low sea

ice concentrations in the Arctic already caused atmospheric circulation anomalies in winter (Lee et al., 2015). Hence transport pathways for aerosol and pollution into the Arctic in general will change due to the changing circulation pattern in association with Arctic amplification.

To date it is well known that transport into the Arctic and especially into the Arctic lower troposphere is possible along different pathways depending on the source area of air masses and the time of the year (Klonecki, 2003; Stohl, 2006; Law

and Stohl, 2007; AMAP, 2015). Stohl (2006) identified three major pathways, which significantly contribute to transport from major pollution sources into the Arctic lower troposphere:

1) Rapid low level transport which is followed by an uplift at the Arctic front at the location and time when the Arctic front is located far north. For this transport route uplift and potential precipitation occurs mostly north of 70°N which allows for significant deposition of aerosol and water-soluble pollutants in the Arctic. In their study they estimated a transport time of

4 days or less. Significant emissions only from densely populated regions in Europe are able to be transported into the high Arctic lower troposphere via this route since major emission regions in North America and Asia are located south of the polar front. Note that the Arctic front and the polar front are geographically two distinct features. The Arctic front, which is best expressed during the summer months, is thought to develop due to strong differential heating between the cold Arctic ocean and adjacent ice and snow free land (Serreze et al., 2001; Crawford and Serreze, 2015). It marks the southern boundary of the

cold Arctic air mass that is separated from the less cold polar air mass at the Arctic front. The polar front in contrast is the well known frontal zone separating warm mid-latitude and subtropical air masses from cold polar air masses. It is in general located further south compared to the Arctic front and displaced in equatorward direction in summer and in poleward direction in winter. In this baroclinic region characterized by strong horizontal temperature gradients cyclones develop from an initial disturbance at the front. During the winter months the Arctic front can extend far south over the continents and can eventually

be co-located or merge with the polar front.

2) Low level transport of already cold air masses into the polar dome, which is associated with further diabatic cooling during the transport time scales of 10-15 days. This pathway from European and high latitude Asian sources mainly occurs during winter, since transport over snow-covered regions (e.g. Siberia) is involved. Thus, strongly polluted air masses could be transported into the high Arctic lower troposphere. This transport pathway is negligible during the summer months when the

surface in Eurasia is a net source of heat (Klonecki, 2003).

3) Fast uplift mainly due to convection in southern mid-latitudes which is then followed by high altitude transport in northerly directions. Radiative cooling eventually leads to a slow descent into the polar dome area after air masses have arrived in the high Arctic. Being less frequent from Europe, this transport pathway is mostly prevalent from North America and East Asia. In contrast to the other two transport pathways, scavenging processes can occur during the strong ascent in mid-latitudes which

can lead to a significant washout of aerosol and soluble pollutants already outside the Arctic.



The high Arctic lower troposphere in general is quite well isolated from the rest of the Arctic by a transport barrier referred to as the polar dome. The polar dome is formed by sloping isentropes, the isolines of potential temperature Θ, as a result of radiative cooling in the high Arctic especially during the winter months without sunlight (Barrie, 1986; Klonecki, 2003; Stohl, 2006). Air masses preferably keep their potential temperatures almost constant during transport, since atmospheric circulation

can be well described by adiabatic motions in the absence of diabatic processes related to clouds, radiation and turbulence. The potential temperature is low within the polar dome area and thus only air masses which experienced diabatic cooling are able to enter the polar dome from specific source regions as discussed before (Stohl, 2006). As a consequence the commonly known "Arctic haze" phenomenon is mainly fed by northern Eurasian pollution sources as those air masses are cold enough to enter the high Arctic lower troposphere (Carlson, 1981; Rahn, 1981; Raatz, 1985; Iversen, 1984; Barrie, 1986; Brock et al.,

1990; Dreiling and Friederich, 1997). Already known for decades Arctic haze has again gained attention at the beginning of the $21^{st}$ century which was mainly triggered by black carbon (BC) and its role in Arctic climate change (Flanner et al., 2007; Hansen and Nazarenko, 2004; Law and Stohl, 2007; McConnell et al., 2007; Quinn et al., 2008; Shindell and Faluvegi, 2009). Pollution originating from outside the Arctic is transported into the high Arctic lower troposphere during the winter months and the lack of sunlight allows for a build-up of aerosol particles and gaseous pollutants: When temperatures during the winter

months become extremely low near the surface the Arctic lower troposphere is thermally very stably stratified accompanied by surface based inversions that can persist for several days (Bradley et al., 1992). Turbulent exchange and hence dry deposition is reduced under these conditions. Furthermore the lower troposphere is extremely dry which prevents scavenging of aerosol and gaseous pollutants by wet deposition. At the spring time peak Arctic haze is often visible as layers of brownish haze affecting the radiation budget of the Arctic lower troposphere and also contributing to contamination of the Arctic environment. During

the transition to pristine summer conditions Arctic haze declines which is mainly caused by efficient aerosol scavenging in mid-latitudes during convective uplift of air masses. Anthropogenic aerosol is further reduced by frequent precipitation of low intensity within the Arctic lower troposphere (Barrie, 1986; Browse et al., 2012; Garrett et al., 2010).

In general, the polar dome boundary acting as a transport barrier for warmer mid-latitude air masses is variable in time and space. Synoptic disturbances can lead to a shift of the polar dome boundary or perturb the transport barrier fostering exchange

with mid-latitude air that can alter the composition of the lower Arctic troposphere. A distinct definition of the polar dome boundary location is crucial to understand and quantify these effects . Although the polar dome feature is known for decades, only very few specific definitions of the polar dome boundary have been described in the literature. Early studies by Carlson (1981) or Raatz (1985) identified the polar front as a transport barrier decoupling the Arctic from influence of mid-latitude air masses for their analyses of Arctic haze. More recent studies used the location of the Arctic front as a marker for the polar

dome as a transport barrier (Klonecki, 2003; Stohl, 2006). Jiao and Flanner (2016) used the maximum zonal mean latitudinal gradient of $500\,\mathrm{hPa}$ geopotential height in the Northern Hemisphere. One of the drawbacks of the latter approach is the missing definition at lower altitudes.

It was previously mentioned that the polar dome is well isolated from the surrounding troposphere. This leads to long residence times of air masses within the polar dome. Anthropogenic tracers like CO and $CO_2$ show temporal changes within

days to weeks due to changes in emissions and thus the source strength of these species. Furthermore the distribution of sources





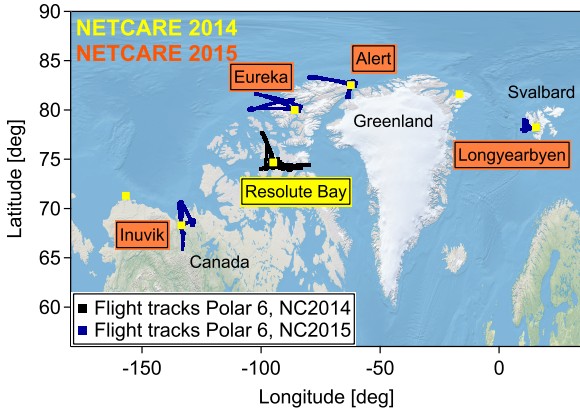

**Figure 1.** Compilation of flight tracks of research flights during two NETCARE airborne field campaigns in July 2014 and April 2015.

and sinks as well as the efficiency of removal processes for both species is different within the Arctic and at mid-latitudes thus leading to latitudinal gradients for both species. Taking into account the isolation of the polar dome, tracer gradients across the polar dome boundary as a transport barrier should establish. In this study we use trace gas gradients of CO and $CO_2$ between inside and outside of the polar dome to derive a tracer based diagnostic to identify the location of the polar dome boundary.

The basis are two airborne field campaigns: NETCARE 2014 in July and NETCARE 2015 in April (Section 2, 3, 4). Thus, the time period of the late spring and summer are covered. Despite focussing on only these specific time periods, this study is the first attempt to define the polar dome boundary based on airborne trace gas gradients (Section 5). Furthermore, an analysis of the transport history of air masses in the polar dome and the surrounding is presented.

## 2 The NETCARE project

The NETCARE project (Network on Climate and Aerosols: Addressing Key Uncertainties in Remote Canadian Environments, http://www.netcare-project.ca) is configured around four research activities addressing key uncertainties in the field of Arctic aerosol climate research (Abbatt et al., 2018). Within this framework two aircraft based measurement campaigns were performed in the high Arctic. The main objectives of both campaigns were to study aerosol-cloud interaction as well as to characterize local and remote sources for pollution within the high Arctic lower troposphere in summer 2014 (July) and spring

2015 (April). Figure 1 shows a compilation of flight tracks for the two airborne research activities named NETCARE 2014 and NETCARE 2015.

The first project was performed from July $4^{th}$ to July $21^{st}$, 2014 with the Polar 6 aircraft based in Resolute Bay, Nunavut, Canada (e.g. Aliabadi et al. (2016); Leaitch et al. (2016); Willis et al. (2016, 2017); Burkart et al. (2017); Köllner et al. (2017)). In total 11 research flights, each between 4-6 hours long, covered two main research areas, the Lancaster Sound east of Resolute

Bay and the area north of Resolute Bay where two polynyas were located. During the last part of the campaign, July $19^{th}$ to July $21^{st}$, a special research focus was on ship emission measurements (Aliabadi et al., 2016).



**Table 1.** Location and coordinates for the different stations from which measurement flights were performed during the two NETCARE airborne projects in July 2014 and April 2015. Additionally the time at the station and the number of research flights are given.

|  | Location | Coordinates | Date | Flights |
|---|---|---|---|---|
| NETCARE 2014 | Resolute Bay | 74.7°N, 95.0°W | July $4^{th}$ to July $21^{st}$ | 11 |
| NETCARE 2015 | Longyearbyen | 78.2°N, 15.5°E | April $5^{th}$ | 1 |
|  | Alert | 82.5°N, 62.3°W | April $7^{th}$ to April $9^{th}$ | 4 |
|  | Eureka | 80.0°N, 85.8°W | April $11^{th}$ to April $17^{th}$ | 2 |
|  | Inuvik | 68.3°N, 133.5°W | April $20^{th}$ to April $21^{st}$ | 3 |

The second aircraft project took place in April, 2015. We performed pan-Arctic measurements throughout the European and Canadian Arctic (see Fig. 1). This campaign was a joint NETCARE and PAMARCMiP (https://www.awi.de/en/science/climate-sciences/sea-ice-physics/projects/netcare-arctic-study-of-short-lived-climate-pollutants/pamarcmip-2015.html) project, which will be referred to as "NETCARE 2015" throughout this paper (e.g. Libois et al. (2016); Willis et al. (2019); Schulz et al.

(2018)). During 10 research flights, each 4-6 hours long, we specifically focused on a better understanding of aerosol transport into the Arctic in early spring and its influence on ice cloud formation. More details of the different base stations can be found in Tab. 1. Multiple vertical profiles from the lowest possible altitude (60 m) up to 6000 m were performed to study the vertical distribution of aerosol particles and trace gases.

## 3 Methodologies

### 3.1 Measurements and data

Airborne measurements were performed using the Polar 6 aircraft of the Alfred Wegener Institute Helmholtz Center for Polar and Marine Research, Bremerhaven, Germany. Polar 6 is a DC 3 aircraft converted to a Basler BT67 (Herber et al., 2008) and modified for operation in cold and harsh environments. The aircraft has a non-pressurized cabin, however flights up to an altitude of 6 km were regularly performed during the 2014 and 2015 campaigns. The typical survey speed of the aircraft is

120 kts ($\cong 60\,\mathrm{m\,s^{-1}}$) with ascent and descent rates of $150 - 300\,\mathrm{m\,min^{-1}}$ during the vertical profiles.

### 3.2 Instrumentation

Meteorological and aircraft altitude data for Polar 6 are provided by the AIMMS-20 instrument. The instrument was designed and manufactured by Aventech Research Inc., Barrie, Ontario, Canada. It includes the Air Data Probe (ADP) that reports the three-dimensional, aircraft-relative flow vector consisting of true air speed, angle-of-attack and sideslip. In the rear section

of the instrument temperature and relative humidity sensors are located providing data with an accuracy of 0.30°C and a resolution of 0.01°C for temperature measurements and 2.0% and 0.1% for humidity measurements, respectively. A GPS module provided the aircraft 3-D position and inertial velocity. Horizontal and vertical wind speeds were measured with





accuracies of 0.50 and $0.75\,\mathrm{m\,s^{-1}}$, respectively. All data were internally sampled with $200\,\mathrm{Hz}$ resolution and for further analysis averaged to $1\,\mathrm{Hz}$ resolution. From the AIMMS-20 data set especially the temperature and pressure data are used throughout this study. The instrumentation for aerosol and cloud droplet as well as upwelling radiance measurements is described in detail in Leaitch et al. (2016); Willis et al. (2016); Burkart et al. (2017); Aliabadi et al. (2016); Libois et al. (2016) and Schulz et al.

5 (2018).

CO was measured with an Aerolaser ultra fast carbon monoxide (CO) monitor model AL 5002 based on VUV-fluorimetry, using the excitation of CO at $150\,\mathrm{nm}$ (Gerbig et al., 1999). UV light stems from a resonance lamp excited by a Radio Frequency (RF) discharge. The selection of the $150\,\mathrm{nm}$ wavelength is realised by an optical filter, which images the lamp into the RF chamber, where fluorescence is viewed at a right angle by means of a photomultiplier tube (PMT) with suprasil optics.

The optical filter consists of two $CaF_2$ lenses. The two dielectric mirrors provide the spectral band path (bandwidth of $8\,\mathrm{nm}$ full width at half maximum (FWHM) at approximately $150\,\mathrm{nm}$). The instrument was modified for applying in-situ calibrations during in-flight operations. These regular in-situ calibrations are performed on a 15 to $30\,\mathrm{min}$ time interval during measurement flights using a NIST traceable calibration gas with a known CO concentration at atmospheric levels as well as zero measurements. Calibrations and zero measurements account for instrument drifts. CO data achieved a precision ($1\,\sigma$, $1\,\mathrm{Hz}$)

of $2.2\,\mathrm{ppbv}$ during NETCARE 2014 and $1.5\,\mathrm{ppbv}$ during NETCARE 2015. The stability of the instrument is calculated to $4.1\,\mathrm{ppbv}$ and $1.7\,\mathrm{ppbv}$, respectively, before applying the post flight data correction. Stability is based on the mean drift between two subsequent calibrations which were performed during flights. Stability is mainly affected by temperature variations. These instrumental drifts are corrected after the flights assuming linear drift. Hence, the total uncertainty relative to the working standard of $4.7\,\mathrm{ppbv}$ for NETCARE 2014 and $2.3\,\mathrm{ppbv}$ for NETCARE 2015 can be regarded as an upper limit.

$CO_2$ was measured with a LI-7200 closed $CO_2/H_2O$ Analyzer from LI-COR Biosciences GmbH. The instrument simultaneously also measures water vapour, which is used for $CO_2$-$H_2O$-interference corrections. The measurement principle is based on an optical source emitting infrared light through a chopper filter wheel and the enclosed sample path to a temperature controlled lead selenide detector. By using the ratio of absorption by carbon dioxide in the sample path to a reference, the density of the gases and thus the mixing ratio can be calculated. The instrument itself was mounted in a 19", 3 HE rack mount includ-

ing additional components for flow control and in-situ calibrations during in-flight operations. As for the CO measurements, calibrations were performed on a regular time interval of 15 to 30 minutes using a NIST traceable calibration gas with a known $CO_2$ concentration at atmospheric levels and a water vapour concentration close to zero. $CO_2$ data during NETCARE 2014 achieved a precision ($1\,\sigma$, $1\,\mathrm{Hz}$) of $0.02\,\mathrm{ppmv}$ and $0.05\,\mathrm{ppmv}$ during NETCARE 2015. Using the same methodology as for CO, the stability of the instrument is calculated to $0.76\,\mathrm{ppmv}$ for NETCARE 2014 and $1.72\,\mathrm{ppmv}$ for NETCARE 2015, before

applying the post flight data correction. The total uncertainty relative to the working standard thus amounts to $0.76\,\mathrm{ppmv}$ for NETCARE 2014 and $1.72\,\mathrm{ppmv}$ for NETCARE 2015. The uncertainty for the measurement of $H_2O$ is $18.5\,\mathrm{ppmv}$ or $2.5\,\%$, whichever is greater.





### 3.3 LAGRANTO backward trajectories

We used the Lagrangian analysis tool (LAGRANTO) (Wernli and Davies, 1997; Sprenger and Wernli, 2015) to determine the origin of air masses that were sampled. LAGRANTO trajectories were calculated based on operational analysis data from the European Centre of Medium-Range Weather Forecasts (ECMWF). This data has a horizontal grid spacing of $0.5°$ with
137 hybrid sigma-pressure levels in the vertical from the surface up to $0.01\,\mathrm{hPa}$. Trajectories were initialized every $10\,\mathrm{s}$ from coordinates along individual research flights and calculated 10 days back in time. The location of the individual trajectory is available at a $1\,\mathrm{h}$ time interval. Different variables of atmospheric state were simulated along the trajectory (temperature, potential temperature, potential vorticity, specific humidity, cloud water and cloud ice water content, Richardson number and equivalent potential temperature).
To account for the latitudinal transport history of the air parcels we calculated the median latitude along the trajectories. We used this as a proxy for the most representative position during the last 10 days associated with the respective values of CO and $CO_2$. To account for diabatic descent occurring during transport we calculated the maximum potential temperature. Both parameters are used in Sec. 5.3 for the analysis of the polar dome boundary.

### 4   Meteorological Overview

#### 4.1   NETCARE 2014

Flights during the NETCARE 2014 field campaign were performed during Arctic summer in July in the area around Resolute Bay, Nunavut, Canada. The meteorological situation can be separated into two different meteorological regimes (see Fig. 2). During the first phase (July $4^{th}$ to July $12^{th}$) the boundary layer was capped at low altitudes by a distinct temperature inversion leading to a very stable stratification of the lower troposphere. The prevailing influence of a high pressure system provided ideal
conditions for aircraft based measurements with mainly clear sky, only few or scattered clouds and low wind speed. Beginning July $13^{th}$ Resolute Bay was influenced by a low pressure system located to the west above the Beaufort Sea. This system eventually passed Resolute Bay two days later. Increased humidity, precipitation and fog characterized the local weather and prevented Polar 6 from flying until July $17^{th}$. The last flights of the campaign were performed between July $19^{th}$ and July $21^{st}$ when a pronounced low pressure system south of Resolute Bay and centred around King William Island influenced the
measurement region (see Fig. 2b). Increased wind speeds, mostly mid to high level clouds and precipitation resulted from the inflow of warm air from more southern latitudes. Furthermore this situation was favourable for mid-latitude air masses being advected to the measurement region potentially affecting concentrations levels of trace gases and aerosol particles.

#### 4.2   NETCARE 2015

The NETCARE 2015 pan-Arctic study was performed during Arctic spring in April in the European and Canadian Arctic. The
aircraft were based at four different locations, namely Longyearbyen (Norway), Alert, Eureka (both Nunavut, Canada) and Inuvik (Northwest Territories, Canada), allowing for a wider coverage of the entire Arctic. Figure 3 shows mean geopotential



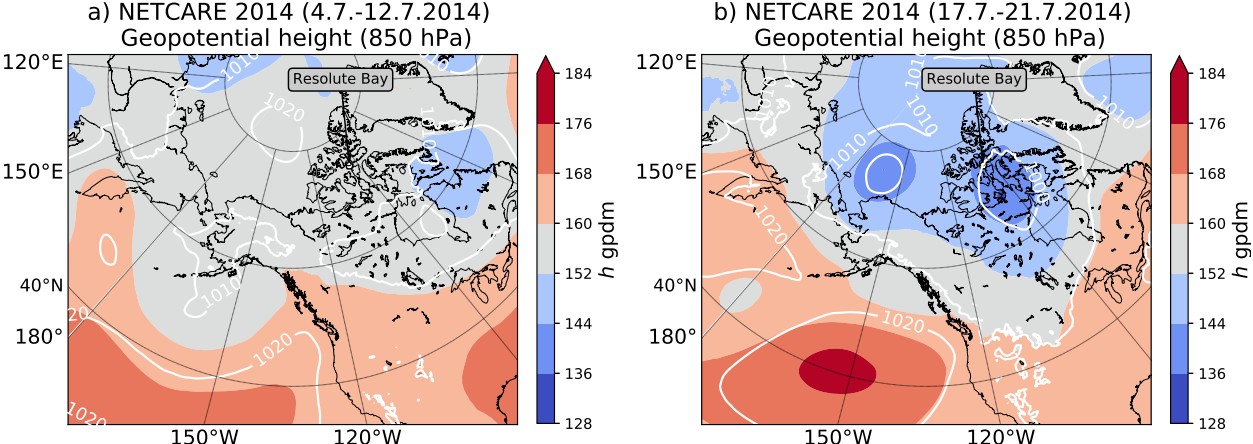

**Figure 2.** Mean geopotential height on $850\,\mathrm{hPa}$ for the period from July $4^{th}$ 2014 to July $12^{th}$ 2014 (a) and for the period from July $17^{th}$ 2014 to July $21^{st}$ 2014 (b).

height at $850\,\mathrm{hPa}$ over the time interval of the measurements in the respective region. At the time of the first flight of the campaign in Longyearbyen (April $5^{th}$, see Fig. 3a), Spitsbergen was under a quite stable high pressure influence with almost no clouds and only weak winds. During the measurements in Alert the meteorological situation was dominated by a pool of cold air centred above Ellesmere Island to the south-west of Alert. A cyclonic flow was established around this cold air

guiding low pressure systems around the cold pool and thus preventing mid-latitude air masses to potentially influence the high Arctic lower troposphere. Stable conditions with almost clear sky facilitated airborne measurements on 4 research flights between April $7^{th}$ and April $9^{th}$ during this period (see Fig. 3b). After the transfer to Eureka on April $10^{th}$ two research flights were performed in almost the same meteorological conditions as in Alert. When the surface low started moving south over Baffin Bay from April $13^{th}$ on (see Fig. 3c) strong northerly and north-easterly winds in the lower troposphere influenced the

measurement regions. The warmer flow was guided from southern areas over open water around the low pressure centre and was associated with moisture transport to the land leading to cloud formation and fog which impeded research flights out of Eureka. In the following days a low pressure system started intensifying north of Greenland and maintained the low level moist northerly flow. The last flights of the campaign were conducted in Inuvik between April $20^{th}$ and April $21^{st}$. After the ferry from Eureka to Inuvik on April $17^{th}$ and $18^{th}$ high pressure influence was prevalent in Inuvik. At the time of the research

flights a low pressure system located over Alaska fostered a southerly and south easterly flow into the Inuvik area favourable for mid-latitude air masses to enter the measurement region (see Fig. 3d).





**Figure 3.** Mean geopotential height on $850\,\mathrm{hPa}$ from April $5^{th}$ 2015 (a), from April $7^{th}$ 2015 to April $9^{th}$ 2015 (b), from April $11^{th}$ 2015 to April $13^{th}$ 2015 (c) and from April $20^{th}$ 2015 to April $21^{st}$ 2015 (d).

## 5 Results

### 5.1 Air mass history

During the flights of the first period of NETCARE 2014 until July $12^{th}$ high pressure influence was prevailing. Air masses tended to stay within the high Arctic and circle around the measurement region. Almost no mid-latitude influence in terms of trajectory origin was observed when using the Arctic circle as a boundary between the Arctic and mid-latitudes. This is evident in Fig. 4a, which shows the area weighted accumulated number of trajectory points per grid box ($1°$ x $1°$) for the last ten





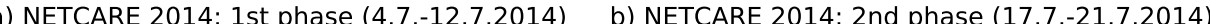

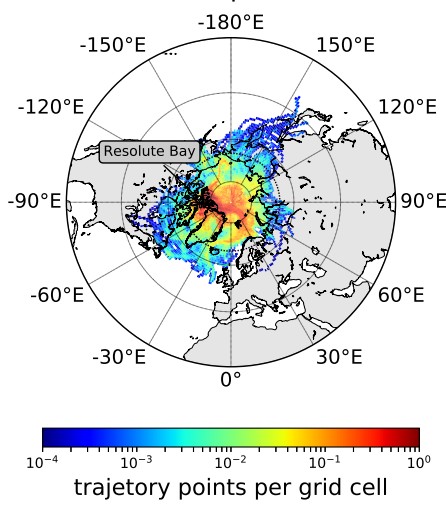
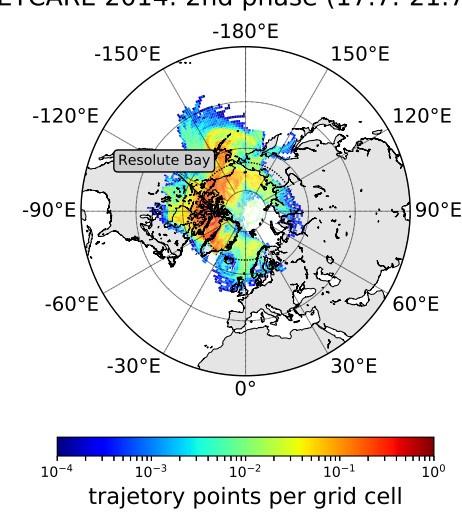

**Figure 4.** Area weighted trajectory density during the different campaign phases in July 2014 (gridded by $1°$ x $1°$). The colour code represents the amount of trajectory points per grid box weighted by the area of the grid box. The individual panels show the results for the first phase (a) and the second phase (b). The bold black dashed circle denotes the Arctic circle.

days before the measurements during the summer campaign in July 2014. The highest density of trajectory points is observed significantly north of the Arctic circle. Note that multiple "hits" of one trajectory in a specific grid box are possible during the 10 day travel of the air mass associated with the trajectory.

In contrast, during the flights within the second period (July $17^{th}$ to July $21^{st}$) more air masses originated in regions south
of the Arctic circle (see Fig. 4b). Highest trajectory densities are found slightly north of the Arctic circle in the Canadian Arctic Archipelago extending southward to continental Canada and the Bering Sea. The stable low pressure system over King William Island thus favours the transport of mid-latitude air masses to the high Arctic, and the potential for a stronger impact of mid-latitudinal sources on the Arctic chemical composition.

During pan-Arctic measurements in April 2015 probed air masses show very different histories depending on the respective
measurement location in the Arctic (see Figs. 5a-d). Measurements performed from the two northernmost stations Alert and Eureka are less influenced by air masses of mid-latitudinal origin than those further south. This indicates that synoptic disturbances did not have a strong influence on the high Arctic stations at least during the one month period, which is covered by our measurements and the backward trajectories. Based on the "density maps" of the trajectory locations, the strongest mid-latitude influence is indicated during the measurements in Inuvik between April $20^{th}$ and April $21^{st}$ (Fig. 5d). During
three flights a warm conveyor belt (WCB) type transport associated with a strong low pressure system over Alaska influenced the tropospheric composition in the measurement region by advecting pollution from South-East-Asia. As already mentioned



**a) NETCARE 2015: Longyearbyen (5.4.2015)**

**b) NETCARE 2015: Alert (7.4.-9.4.2015)**

**c) NETCARE 2015: Eureka (11.4.-13.4.2015)**

**d) NETCARE 2015: Inuvik (20.4.-21.4.2015)**

**Figure 5.** Area weighted trajectory density during the different campaign locations in April 2015 (gridded by $1°$ x $1°$). The colour code represents the amount of trajectory points per grid box weighted by the area of the grid box. The individual panels show the results for the measurements in Longyearbyen (a), Alert (b), Eureka (c) and Inuvik (d). The bold black dashed circle denotes the Arctic circle.

before the high Arctic stations in Eureka and Alert were much less affected by air masses from mid-latitudes, which is confirmed by the analysis of the air mass history for the flights between April $7^{th}$ and April $13^{th}$ (Figs. 5b and c) when only a few trajectories travel over areas outside the Arctic circle. Air masses mainly resided over the Canadian Arctic Archipelago and





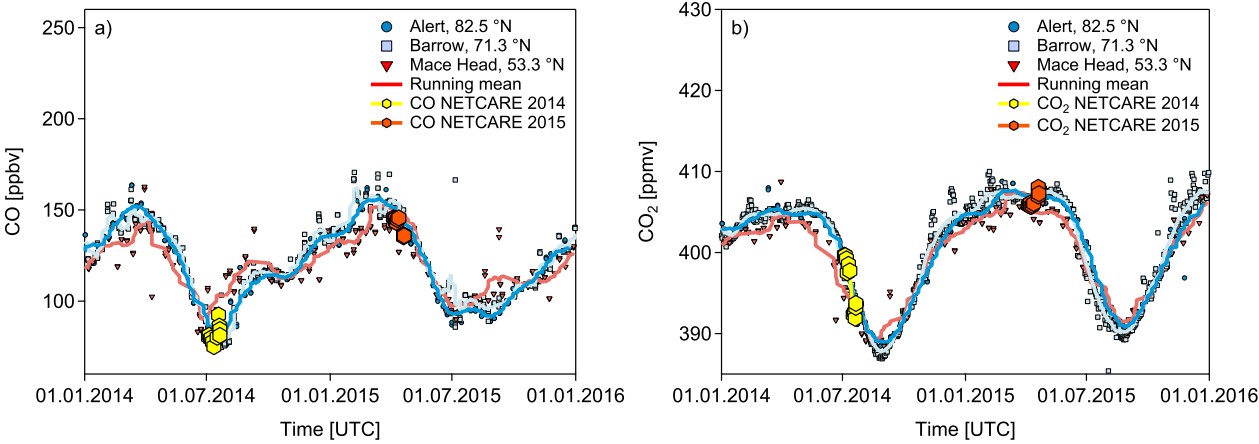

**Figure 6.** CO (a) and $CO_2$ (b) seasonal cycle based on NOAA ground based measurements in Alert (Canada), Barrow (Alaska) and Mace Head (Ireland) for the years 2014 and 2015. The running mean line colours correspond to those of the station symbols. The aircraft data of the lowest 200 m are overlayed as mean plus standard deviation for the respective flight. Error bars for the aircraft data are too small to be visible. NETCARE 2014 data are in yellow and NETCARE 2015 data are in orange.

northern Greenland with only episodic influence from the North American continent and Siberia. In Longyearbyen only one flight was performed on April $5^{th}$, which shows a mixture of mid-latitude and Arctic air masses (Fig. 5a). The origin of the majority of air masses contributing to the observations in Longyearbyen was in northern and eastern parts of Europe.

### 5.2 Trace gas observations

According to the meteorological situation and the general transport regimes a significant influence of the air mass history on the mixing ratios particularly of CO and $CO_2$ is expected. Both species show latitudinal and vertical gradients (see Fig. 6) and in addition, both are affected by anthropogenic pollution which makes them ideally suited to identify pollution events affecting the Arctic background. Furthermore, both species show a seasonal cycle in the Arctic.

Figure 6 shows the ground based observations of CO (Dlugokencky et al., 2018) (a) and $CO_2$ (Petron et al., 2018) (b) for
three relevant sites, namely Alert (Canada, $82.4°$ N), Barrow (Alaska, $71.3°$ N) and Mace Head (Ireland, $53.3°$ N) for the years 2014 and 2015. Superimposed are the respective trace gas measurements of CO and $CO_2$ for altitudes below 200 m as mean values for each individual flight during the two NETCARE campaigns in July 2014 and April 2015. For CO and $CO_2$ aircraft based and ground based observations show a very good agreement. In April 2015 trace gas levels are in general higher compared to July 2014. The observed change in trace gas levels between spring and summer reflects the typical seasonal cycle
of these two species in the Arctic.

For CO the seasonal cycle in the high Arctic shows a maximum in late winter/early spring and a minimum during late summer. This seasonal cycle maximum reflects quite well the transport of anthropogenic pollutants - mainly from fossil fuel burning - from northern Europe and Siberia into the high Arctic lower troposphere during winter (Klonecki, 2003; Stohl,



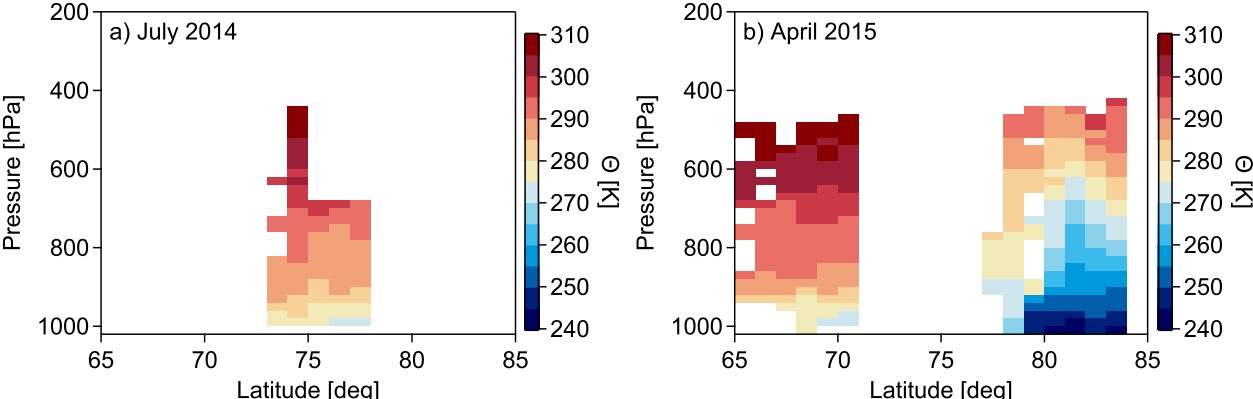

**Figure 7.** Potential temperature $\Theta$ as a function of latitude and pressure binned in steps of $1°$ latitude and $20\,\text{hPa}$ pressure for July 2014 (a) and April 2015 (b). Note the dome like structure during April 2015 (NETCARE 2015) which is virtually absent for the Resolute Bay data during July 2014 (NETCARE 2014).

2006). Since photochemically produced OH is absent during wintertime due to the lack of sunlight, CO in the high Arctic has no significant sink which results in a longer chemical lifetime of CO in the order of months. Hence CO increases over the course of the winter, in particular within the polar dome (Novelli et al., 1998; Engvall et al., 2008). As soon as the sunlight returns during late February and early March there is a sharp transition between 24 h polar night and 24 h polar day. The

increasing concentration of OH leads to increased oxidation of CO and a shorter lifetime in the order of weeks (Dianov-Klokov and Yurganov, 1989; Holloway et al., 2000). During the transition from spring to summer (April to June) the photochemical activity in the Arctic and smaller emissions of CO in mid-latitudes lead to decreasing CO in the Arctic until the minimum is reached at the end of the summer (Barrie, 1986; Klonecki, 2003; Engvall et al., 2008).

The seasonal cycle for $CO_2$ in the northern hemisphere is mainly controlled by carbon uptake and release processes of

the biosphere (Keeling et al., 1996; Forkel et al., 2016). Whereas during the summer months the $CO_2$ seasonal cycle reaches its minimum due to photosynthetic carbon uptake by vegetation, respiration of the biosphere is prevalent during wintertime. Particularly in Arctic winter, the absence of sunlight allows for a build-up of $CO_2$ concentrations. However, meridional $CO_2$ transport into the high Arctic by synoptic weather disturbances plays a critical role for the seasonal cycle there and dominates over local atmosphere-biosphere fluxes (Fung et al., 1983; Parazoo et al., 2011; Barnes et al., 2016). As a result the synoptic

eddy driven meridional transport reduces the seasonal cycle in mid-latitudes and amplifies it in polar regions leading to a meridional $CO_2$ gradient (Parazoo et al., 2011).

### 5.3  The location of the polar dome

Conceptually, the polar dome can be regarded as the region below the upward sloping isentropes north of the region where low value isentropes intersect the surface. Figure 7 shows the observed potential temperature ($\Theta$) distribution as a zonal mean

for the respective campaigns. Potential temperature was calculated from temperature and pressure measurements on board





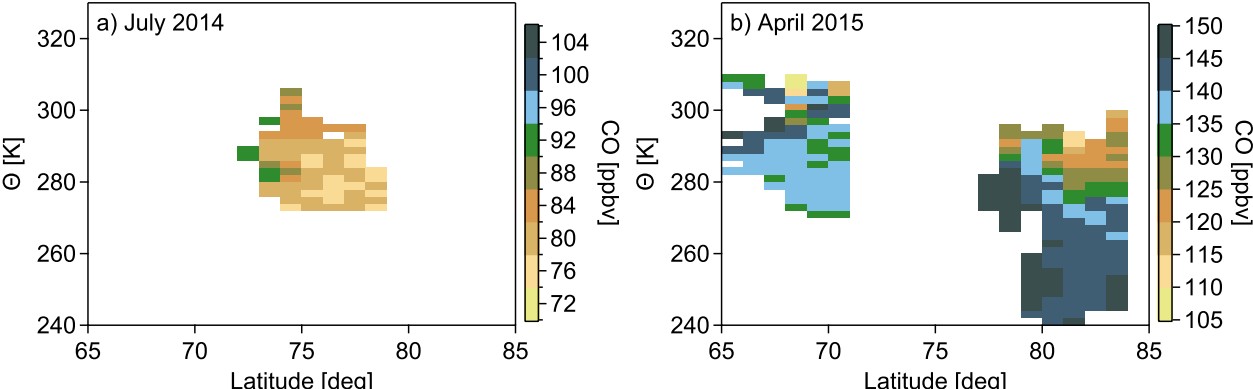

**Figure 8.** CO distribution binned by latitude and potential temperature for July 2014 (a) and April 2015 (b). The colour code represents the average CO mixing ratio calculated from all data points in the respective $1°$ latitude and $2\,K$ bin interval. Note that only background mixing ratios are shown now. Polluted air masses are identified and filtered when the average background distribution is exceeded by $2\sigma$.

the Polar 6 aircraft. A dome-like structure of the isentropes is visible for April 2015 (NETCARE 2015). Minimum potential temperatures lower than $275\,K$ were only present in the high Arctic lower troposphere north of $70°N$. In contrast, a dome-like structure is hardly visible for July 2014 (NETCARE 2014). Only below $950\,hPa$ and north of $75°N$ $\Theta$ values of $275\,K$ were observed. This is in agreement with previous studies which showed that during the summer months the extent of the polar

dome is much smaller compared to the winter time (Klonecki, 2003; Stohl, 2006; Jiao and Flanner, 2016).

If we now use potential temperature as the vertical coordinate, air masses within the polar dome associated with the coldest potential temperatures should separate from other regions. This is evident on Fig. 8, in particular for the April 2015 measurements (b). The colour code in Figs. 8a and b represents the average CO background mixing ratio calculated from all data points within the respective bin interval. For the polar dome analysis we only use background trace gas mixing ratios. We exclude

polluted air masses, if the mean background CO mixing ratio is exceeded by 2 standard deviations. In April 2015 (Fig. 8b) northernmost latitudes exhibit the largest CO values of 140-150 ppbv for potential temperatures lower than $275\,K$. At higher isentropes typical CO values range from 100-135 ppbv. There, the larger variability of CO mixing ratios indicates different source regions contributing to the observations, which is in particular observed for the lower latitudes. In the distribution for the summer campaign in July 2014 (Fig. 8a) a region of rather uniform low CO is evident north of $75°N$ and below $290\,K$.

Mixing ratios south of $75°N$ and above $290\,K$ tend to be more variable and in general larger than within the aforementioned region. Again the increased variability results from different air mass origins associated with different levels of CO. For both measurement campaigns a distinct transition between the northernmost lower troposphere and regions with lower latitudes and larger potential temperatures is observed. This transition thus indicates a transport barrier for air masses to reach the high Arctic lower troposphere. We hypothesize that those regions north of $75°N$ showing the lowest CO mixing ratios during July

2014 and the largest during April 2015 represent the polar dome whereas the rest of the measurements were collected outside the polar dome.



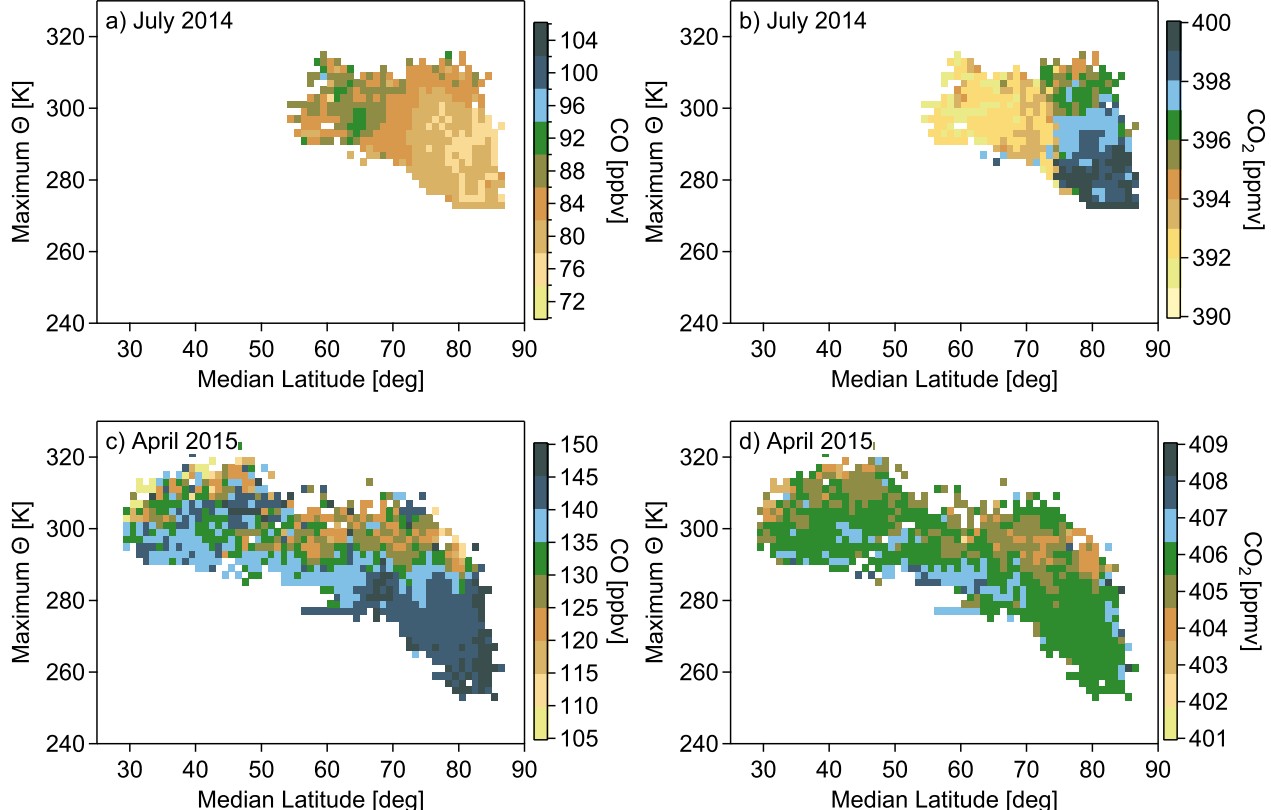

**Figure 9.** (a-d): Trace gas distribution in the maximum potential temperature and median latitude coordinate system. The median and maximum values were derived from every 10-day-trajectory calculated along the flight track. The colour code is representative for the average CO ($CO_2$) value calculated from all data points within one bin interval. The trace gas mixing ratio is the measured value which is assumed to stay constant along the respective trajectory for each measured data point every 10 s.

The trace gas distribution in Fig. 8 only shows a snapshot of the actual situation at the time and the location of the measurement. We used ten day backward trajectories to take into account different transport pathways to the Arctic and the residence times of air masses inside the polar dome area, which is expected to be potentially higher in that region. We display the CO and $CO_2$ distribution in a maximum potential temperature - median latitude coordinate system. Maximum potential temperature and median latitude were derived along every individual trajectory. Median latitude allows for a separation between high Arctic air masses and air masses from mid-latitudes. Air parcels isolated in the polar dome region should stay at high median latitudes whereas air masses extending over a larger meridional distance thus show a more southern median latitude. Maximum potential temperature further allows to account for diabatic descent that air masses experienced during transport. In the maximum potential temperature - median latitude coordinate system those air masses inside the polar dome region exhibit lowest maximum potential temperatures and at the same time largest median latitudes and thus separate from air masses outside the polar



dome. Furthermore, the polar dome is dynamically well isolated from the surrounding troposphere as discussed earlier. Hence, air parcels inside the polar dome are in general not affected by strong mid-latitude CO sources and should show a relatively small CO variability. Therefore, we remapped the CO data to the median latitude and maximum potential temperature along the trajectory to identify transport regimes and the effect of the transport barrier at the polar dome (see Figs. 9a and c). The

majority of trajectories with relatively low CO mixing ratios are confined by the 305 K isentrope and 70 °N as evident in Fig. 9a for July 2014. For April 2015 (Fig. 9c) an area with relatively higher CO is located north of a latitude of 65 °N and below a potential temperature of around 280 K. These findings are supported by the $CO_2$ distribution in the same coordinate system displayed in Figs. 9b and d, which show similar boundaries. Hence, gradients of CO and $CO_2$ establish at these boundaries. In this study these chemical gradients are in particular used to derive a tracer based definition of the polar dome boundary for

the two NETCARE measurement campaigns during July 2014 and April 2015. The determination of the location of the polar dome boundary based on trace gas gradients is discussed in the following section.

**Trace gas gradients**

In July 2014 both species CO and $CO_2$ show a latitudinal gradient across the isentropes for maximum potential temperature levels of $\Theta < 305$ K (see Fig. 9a and b). In particular $CO_2$ shows a strong increase from values around 393 ppmv to 398 ppmv

towards high latitudes in the median latitude range of 70 °N to 75 °N. For CO a decrease from about 83 ppbv to 78 ppbv in this latitude range is also evident. In contrast to $CO_2$ the large variability of CO at lower median latitudes reflects a larger variability of potential source regions, which in turn partly masks the CO gradient. Above 305 K trace gas gradients are weak or absent, indicating rapid isentropic mixing from lower latitudes. We now calculated isentropic trace gas gradients in layers of 2 K for the maximum potential temperature as the vertical coordinate in Fig. 9 to derive the horizontal polar dome boundary.

Hence, for every 2 K altitude interval we determine the latitude of the strongest trace gas gradient. Below 305 K isentropic trace gas gradients maximize around 73 °N. We finally used the median of these maximum gradient latitudes to define the polar dome boundary. If we derive a different median value for the maximum gradient for each of the two species CO and $CO_2$, we consider this difference as the range of the polar dome boundary which can in turn be interpreted as a transition zone rather than a sharp boundary between inside and outside of the polar dome. For July 2014 the average horizontal polar dome

boundary is a sharp transition at 73.5 °N (blue bar in Fig. 10b). The interquartile range denotes to 72.5 °N - 77 °N.

    The strongest vertical gradients of CO and $CO_2$ were determined at maximum potential potential temperature values of 299 - 303.5 K (blue bar in Fig. 10a; interquartile range: 297 to 304.5 K). These values for the upper polar dome boundary are relatively high given a surface value of potential temperature of typically 280 K in summer. A close inspection of the $CO_2$ distribution north of 73.5 °N (Fig. 9b) reveals two layers in the high Arctic separated by approximately 285 K. The vertical

profile of $CO_2$ clearly show the two layers (Fig. 10a). The fact that the vertical profile of CO does not show a clear separation, indicates that rather pristine air masses dominate both layers, which have not experienced strong pollution impact, but rather biogenic impact mainly affecting $CO_2$. If we additionally use this information, we can separate three distinct air masses. The region with lowest potential temperatures ($\Theta < 285$ K) has small (large) mixing ratios of CO ($CO_2$) and is mostly isolated form mid-latitude influence. These air masses are most likely remnants of the spring time polar dome and we refer to this as



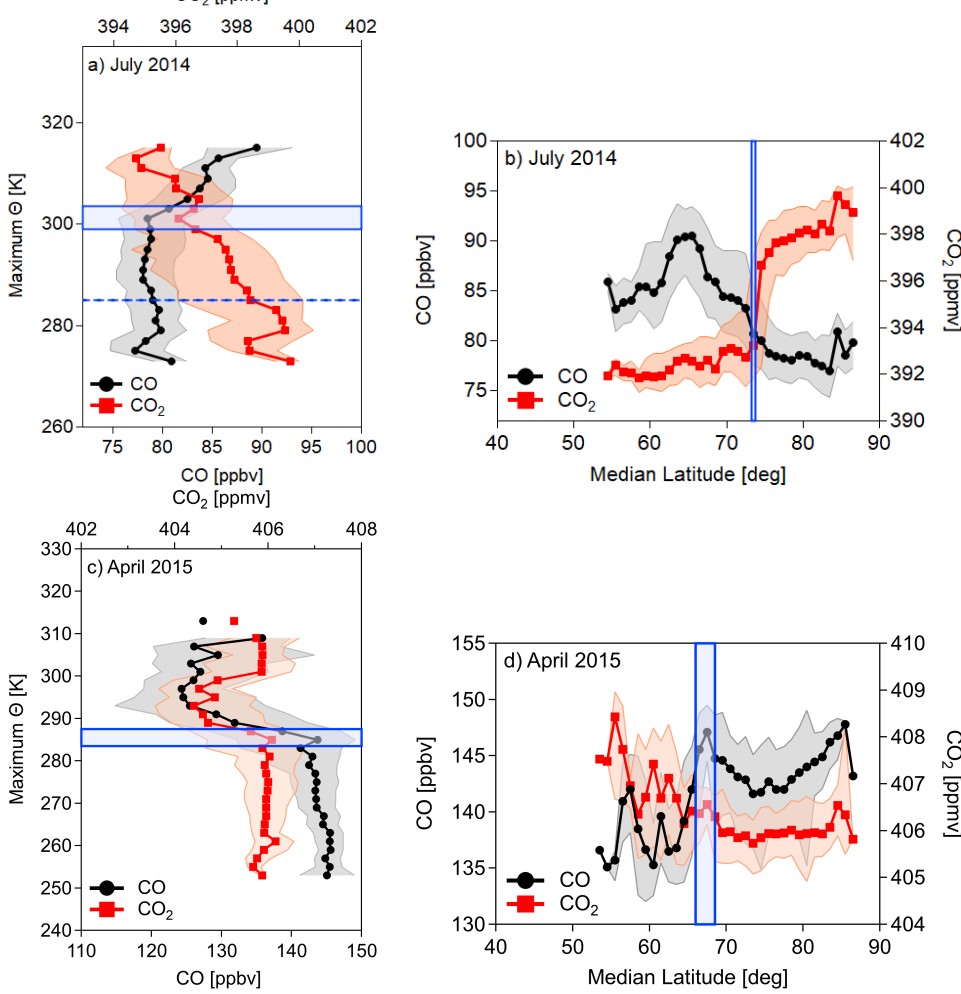

**Figure 10.** Medians of CO and $CO_2$ as a function of median trajectory latitude and maximum trajectory potential temperature (details see text). Median values for the vertical profiles were only calculated north of $75\,°N$ for July 2014 (a) and north of $65\,°N$ for April 2015 (c) because latitudinal gradients indicate a dome boundary north of these median latitudes. At lower latitudes transport and mixing homogenize these gradients (see Fig. 9). For both NETCARE campaigns the median horizontal values were derived only below $300\,K$ (b and d). The blue bar marks the latitude and the potential temperature interval of the strongest change in the tracer mixing ratio, which is interpreted as the transition zone of the polar dome boundary. The shaded area in all figures represents the $1\,\sigma$ standard deviation.

the aged polar dome. Between $285\,K$ and $299\,K$ the air masses still show signatures of the polar dome while also the influence from mid-latitudes increases, indicated by lower $CO_2$ mixing ratios. This region is capped in the vertical by the polar dome boundary spanning from 299 to 303.5 K. Above and thus outside the polar dome mixing ratios of both species clearly show characteristics of mid-latitude influence. Similar values to those observed outside the polar dome were also found in the mid-



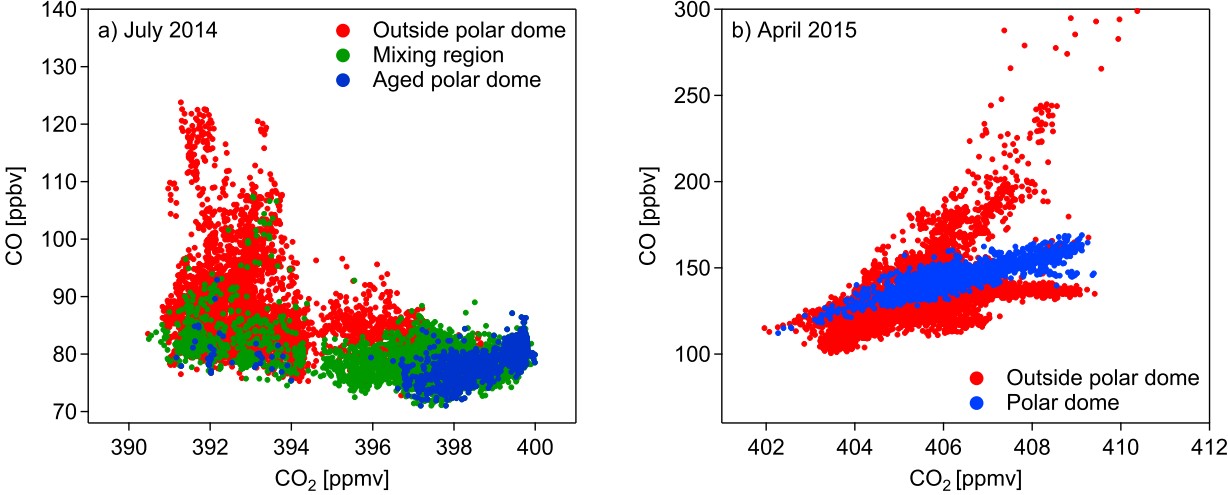

**Figure 11.** (a): Tracer-tracer scatter plot of all data points (background + pollution plumes) within the aged polar dome (blue), the mixing region (green) and outside the polar dome (red) for July 2014. (b): Tracer-tracer scatter plot of all data points (background + pollution plumes) within (blue) and outside (red) the polar dome for April 2015. To separate the different regions the tracer derived polar dome boundaries are used. Boundary values for each region are summarized in Tab. 2

**Table 2.** Maximum potential temperature and median latitude values for the polar dome boundary. Included are also the boundary values used for separating the different regions identified for further analysis.

|  | Maximum potential temperature | Median latitude |
|---|---|---|
| July 2014, polar dome boundary | 299.0 - 303.5 K | 73.5°N |
| April 2015, polar dome boundary | 283.5 and 287.5 K | 66.0°N - 68.5°N |
| July 2014, aged polar dome | $\Theta_{max} < 285.0\,\mathrm{K}$ | $Lat_{med} > 73.5°N$ |
| July 2014, mixing region | $285.0\,\mathrm{K} < \Theta_{max} < 299.0\,\mathrm{K}$ | $Lat_{med} > 73.5°N$ |
| July 2014, outside polar dome | $\Theta_{max} > 303.5\,\mathrm{K}$ | $Lat_{med} < 73.5°N$ |
| April 2015, polar dome | $\Theta_{max} < 283.5\,\mathrm{K}$ | $Lat_{med} > 68.5°N$ |
| April 2015, outside polar dome | $\Theta_{max} > 287.5\,\mathrm{K}$ | $Lat_{med} < 66.0°N$ |

latitude lower troposphere for example at Mace Head observatory in Ireland (see Fig. 6). A summary of the values for the polar dome boundary and the boundaries of the three different regions for July 2014 can be found in Tab. 2.

The threefold structure of the high Arctic lower troposphere based on the derived boundary values for each region summarized in Tab. 2 is further evident in the $CO$-$CO_2$ tracer-tracer correlation in Fig.11a. More precisely, the aged polar dome (blue dots) seems to be a subset of the mixing region (green dots) indicated by a narrow group of data points at end of the highest $CO_2$ and lowest $CO$ mixing ratios. The aged polar dome region is furthermore clearly separated from the region outside (red





dots). The green dots indicate the influence of mixing between dome air and extra-dome air and correspond to the mixing region in the high Arctic between 285 and 299 K (compare Fig. 9b).

The tracer derived polar dome boundary for the April 2015 measurements was on average determined between 66.0 °N and 68.5 °N (blue bar in Fig. 10d; interquartile range: 65.0 °N - 69.5 °N) for the latitudinal value and between potential temperatures
of 283.5 and 287.5 K (blue bar in Fig. 10c; interquartile range: 280.5 and 291.5 K). Values for the polar dome boundary are also summarized in Tab. 2 for April 2015. During spring the $CO$-$CO_2$ tracer-tracer correlation in Fig. 11b indicates at least three distinct branches. The separation of air masses between inside the polar dome and outside is based on the tracer derived polar dome boundary (see Tab. 2). The red branch with the highest CO and $CO_2$ mixing ratios can be associated to pollution events observed during flights in Inuvik. In contrast, the red branch with highest $CO_2$ but relatively low CO values corresponds to
observations in the unpolluted lower troposphere in the Inuvik region. Both branches are clearly associated to air masses outside the polar dome, since measurements around Inuvik were mostly performed outside the determined polar dome boundary. In contrast the blue branch represents the measurements inside the polar dome. These data points show different slopes indicating different air mass properties. Within the polar dome region we observe a mixture of air masses which is evident by the relatively broad range of $CO_2$ and CO values forming a mixing line in Fig. 11b. The lowest CO and $CO_2$ values inside the polar dome
(blue) can be associated to the lowest maximum potential temperatures and thus the highest residence time within the dome area. Air masses with highest CO and $CO_2$ mixing ratios but still inside the polar dome (blue) originate at lower latitudes. In fact, ground stations in the potential source region two weeks before the time of measurement campaign show enhanced CO and $CO_2$ values in the range of the upper branch of the scatter plot of those data points inside the polar dome. The observed mixture of air masses is also reported by Willis et al. (2019) who observed an altitude dependent composition and degree of processing
of aerosol in the spring time polar dome. In their study FLEXPART simulations suggest more southern source regions for those air masses with the highest potential temperatures within the polar dome. Furthermore, Schulz et al. (2018) determined an increase of refractive black carbon (rBC) and a decrease of the rBC mass-mean diameter with potential temperature inside the spring time polar dome, which was also associated to different source regions contributing to the observations.

## 5.4  Air mass statistics and tropospheric composition

Using the tracer-derived polar dome boundary we compare the composition of air masses within the polar dome region and the surrounding (see Tab. 2 for further details on the boundary values). The comparison is based on probability density functions (PDFs) of measured trace gases CO and $CO_2$ for July 2014 (see Figs. 12a and b) and April 2015 (see Figs. 12c and d).

For July 2014 the three different regions identified in Sec. 5.3 are confirmed by the respective PDFs for both species. The aged polar dome and the mixing region show a quite similar distribution for both species except differences in the mode of the
PDF, but are well separated from data outside the polar dome area. Whereas the absolute CO value inside the aged polar dome and the mixing region is lower compared to the area outside the polar dome, the $CO_2$ average mixing ratio within the polar dome is higher compared to the surrounding as summarized in Table 3. This finding can be explained by the seasonal cycle of these two species and their zonal gradients (see data from NOAA ground based measurements in Fig. 6b). The minimum of the seasonal cycle of $CO_2$ in the Arctic and the mid-latitudes is reached at the end of the summer typically during September.


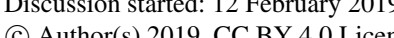

**Figure 12.** Probability density functions (PDFs) of all CO (a and c) and $CO_2$ (b and d) background measurements during July 2014 (upper panel) and April 2015 (lower panel). The median latitude and maximum potential temperature coordinates along the 10 day back trajectories were used for the separation between inside and outside the polar dome using the tracer derived polar dome boundaries (see Tab. 2).

However, the onset of carbon uptake by vegetation in the mid-latitudes starts earlier compared to the high Arctic where less vegetation is prevalent. At the same time the overall burden of $CO_2$ in the Arctic lower troposphere is to a large extent controlled by transport processes (Fung et al., 1983; Parazoo et al., 2011; Barnes et al., 2016). Mid-latitude air with relatively lower $CO_2$ mixing ratios is transported to the high Arctic, in particular during the second phase of the campaign. But, as the polar dome acts as a transport barrier for those air masses, exchange of high Arctic lower tropospheric air with mid-latitude air is reduced leading to the observed PDF for $CO_2$. Under 24 h daylight conditions and with only a few inner Arctic sources of pollution CO concentrations reach their minimum in the high Arctic in late summer. Air masses transported into the high Arctic from more





**Table 3.** Mean and median mixing ratios of CO and $CO_2$ inside and outside of the polar dome area using the tracer derived polar dome boundaries. The respective mixing ratios were calculated based on the minimum latitude and maximum potential temperature coordinates. Note that for the July 2014 dataset also the mixing ratios of the mixing region (MR) are included in the table.

| | CO [ppbv] | | $CO_2$ [ppmv] | |
| | inside polar dome | outside polar dome | inside polar dome | outside polar dome |
| | mean ± sdev (median) | mean ± sdev (median) | mean ± sdev (median) | mean ± sdev (median) |
|---|---|---|---|---|
| July 2014 | 78.8 ± 2.7 (79.1) | 87.6 ± 7.2 (86.0) | 398.6 ± 1.5 (399.0) | 393.1 ± 1.6 (392.8) |
| July 2014, MR | 78.8 ± 3.3 (78.5) | | 397.2 ± 2.0 (397.7) | |
| April 2015 | 142.9 ± 4.2 (143.6) | 133.0 ± 9.9 (134.6) | 406.0 ± 0.6 (405.9) | 405.6 ± 1.1 (405.8) |

southern regions are expected to have relatively higher CO mixing ratios due to the seasonal cycle of CO that has a stronger amplitude in the Arctic compared to mid-latitudes (see Fig. 6a). As the second half of the campaign was dominated by more mid-latitude influence, those air masses enhance the tropospheric CO burden compared to inside the polar dome area which is dominated by photochemically aged low CO air during July 2014.

A strong link between the change in the synoptic situation from a more high pressure controlled regime to a synoptically active regime and the trace gas distributions was observed during the two distinct campaign phases of NETCARE 2014. Based on the trajectory simulations increased mid-latitude influence was observed which in turn influenced the general concentration level of the trace gases CO and $CO_2$ (see Sec. 5.1). The fraction of trajectory points outside the polar dome as a proxy for mid-latitude influence increased from 37 % to 79,% 10 days before the measurements. This in turn led to an increase in the CO

level from 77.9 ± 2.5 ppbv to 84.9 ± 4.7 ppbv. At the same time $CO_2$ decreased from 398.2 ± 1.0 ppmv to 393.8 ± 2.3 ppmv. Furthermore an enhanced variability of CO and $CO_2$ is observed indicating enhanced entrainment of polluted mid-latitude air masses into the high Arctic. Part of the trajectories originating from outside the polar dome area pass the Northwestern Territories at low altitude potentially within the boundary layer where extensive biomass burning was observed during the time of the measurements and before. Accordingly, increased aerosol concentrations during the second half of the campaign were

reported by Burkart et al. (2017).

Using the boundary values listed in Tab. 2 to separate air masses within the polar dome from those outside Fig. 12 shows probability density functions for all CO (c) and $CO_2$ (d) background measurements during April 2015. Based on the PDFs the difference in the tropospheric trace gas composition within the polar dome region and the surrounding is clearly visible for CO but not as distinct as for the July 2014 data set for $CO_2$. Average values for both species in the respective regions

are also summarized in Tab. 3. In general, the distributions of CO and $CO_2$ are much narrower within the polar dome region and CO mixing ratios tend to be higher within the polar dome. For $CO_2$ the mean mixing ratio is quite similar within and outside the polar dome. However, a difference in the general distribution is observed. The reason for a less distinct separation between inside and outside the polar dome area is indicated by the seasonal cycles shown in Figs. 6a and b. $CO_2$ concentration levels are on a plateau with reduced concentration changes with time accompanied by reduced latitudinal gradients. The CO



mixing ratio already started to decrease at the time of the measurements. However, the extent of the polar dome is much larger during the winter months and spring compared to the summer months. Hence more influence from northern mid-latitudes and thus in turn more mid-latitude pollution sources are expected inside the dome area since CO rich air masses originate in cold regions of Eurasia and are able to reach the high Arctic lower troposphere where they are trapped during winter and

early spring (Klonecki, 2003; Stohl, 2006; Jiao and Flanner, 2016). This leads to relatively larger CO levels inside the polar dome compared to the region outside. Air masses from more southern mid-latitudes that are transported above the polar dome, already experienced photochemical loss of CO in their source region.

## 5.5 Transport regimes

In order to analyse the processes dominating the recent transport history of observed air masses during July 2014 and April

2015, we apply the phase-space diagram introduced by Binder et al. (2017). This requires determining the maximum change in potential temperature ($\Delta\Theta$) as the difference between the potential temperature at the time of the measurement ($\Theta_0$) and the previous potential temperature minimum or maximum ($\Theta_{min}$, $\Theta_{max}$) along the trajectory. Depending on which difference is larger in absolute numbers, the air mass has either experienced diabatic heating ($\Delta\Theta = (\Theta_0 - \Theta_{min}) > 0$) or cooling ($\Delta\Theta = (\Theta_0 - \Theta_{max}) < 0$). An analogue analysis is made for the absolute temperature to determine if an air parcel dominantly gained

or lost temperature recently before the measurement. This analysis allows us to cluster the data into four categories shown in Figs. 13a-d and 14a-c. The changes of potential temperature and temperature along the trajectories, which is indicated by the clusters, can be associated with processes affecting the respective air mass. Sector 1 ($\Delta\Theta < 0$, $\Delta T < 0$) mainly contains air masses which experienced diabatic cooling, which indicates either thermal radiation, evaporation or low level transport over snow or ice covered regions and thus cold surfaces. In sector 2 ($\Delta\Theta > 0$, $\Delta T < 0$) air masses gained potential temperature which

indicates an ascending air mass that is diabatically heated by for example solar radiation or condensation processes. Sector 3 ($\Delta\Theta > 0$, $\Delta T > 0$) includes those air masses that experienced both an increase in temperature and potential temperature probably due to solar insolation. Finally, sector 4 ($\Delta\Theta < 0$, $\Delta T > 0$) combines air masses that lost potential temperature and gained temperature during transport. Those air masses are diabatically cooled and thus experience a descent. We apply this clustering approach to identify differences between observations in the different regions (polar dome, outside polar dome, etc.))

and whether we can connect trace gas mixing ratios to the dominant process in a specific region. The different regions were separated using the boundary values listed in Tab. 2, which are based on the tracer derived polar dome boundary.

For July 2014, three regions are of particular interest, (1) the aged polar dome, (2) the mixing region and (3) the region outside the polar dome (see Figs. 13a-d). Within the aged polar dome sector 3 dominates, thus solar insolation is of significant importance to heat the lowest levels. Air masses residing within the lowest altitude experience diabatic heating potentially

resulting in a slow and shallow convective lift of the air masses prior to the time of the measurement. Equal contributions come from sectors 1 and 4, dominated by diabatic cooling either through descent, low level transport over cold surfaces or evaporation. Almost none of the air masses are in sector 2, thus a significant ascent of air masses within the aged polar dome hardly occurs. In contrast, outside the polar dome area sector 4 dominates and generally diabatic cooling occurs (sector 1 and 4). Thus within the polar dome local surface-near diabatic processes seem to mostly affect the air masses which also has an







**Figure 13.** Phase-space diagram illustrating the maximum absolute change in temperature ($\Delta T$) and potential temperature ($\Delta \Theta$) relative to the time of the measurement for July 2014. The colour code denotes the $CO_2$ mixing ratio at the time of the measurement. (a) shows all background data, (b) shows only those data corresponding to the aged polar dome. (c) shows the data points within the mixing region whereas (d) includes all data points outside the polar dome. To separate the different regions the tracer derived polar dome boundaries are used (see Tab. 2).

impact on the chemical composition. Observed $CO_2$ mixing ratios are highest in the aged polar dome associated with aged Arctic air and negligible mid-latitude influence. Outside, air masses have been transported into the Arctic and started to descent caused by radiative cooling. Associated $CO_2$ mixing ratios of these air masses are significantly lower and can be attributed to more mid-latitude regions. Within the dominating sector 4 air masses experience descent once they have reached the high

5 Arctic at higher altitudes. This can be regarded as a typical transport pathway during the summer with a fast uplift of air masses







**Figure 14.** Phase-space diagram illustrating the maximum absolute change in temperature ($\Delta T$) and potential temperature ($\Delta \Theta$) relative to the time of the measurement for April 2015. The colour code denotes the CO mixing ratio at the time of the measurement. (a) shows all background data, (b) only shows those data corresponding to the polar dome and (c) includes all data points outside the polar dome. To separate the different regions the tracer derived polar dome boundaries are used (see Tab. 2).

at mid-latitudes within convective and frontal systems followed by a northward movement and finally a descent into the high Arctic lower troposphere. In between, air masses in the mixing region are even more dominated by sector 4. Within this mixing region trace gas concentrations still show dome-like characteristics thus low level processes seem to dominate over episodes of mid-latitude transport associated with air masses with relatively lower $CO_2$ concentrations. Results for CO confirm the derived

5   transport history for July 2014.





In April 2015 the picture is quite different (see Figs. 14a-c). Inside the polar dome diabatic cooling dominates, in particular sector 1. The reason is diabatic descent due to radiative cooling in the absence of sunlight. In addition low level transport over cold surfaces significantly contributes to transport into the high Arctic. Associated mixing ratios of CO show rather large values that can be explained by the accumulation of anthropogenic pollution from inner Arctic and high northern mid-latitude sources during the winter months. At that time chemistry is reduced in the absence of sunlight leading to an increase in the CO atmospheric lifetime. Air masses within the polar dome are quite efficiently isolated from any significant southern mid-latitude influence. In contrast, outside the polar dome the picture is more diverse. Sector 4 dominates, thus diabatically cooled air masses potentially associated with the tendency to descent are observed. Input from various remote sources leads to a stronger varying CO mixing ratio. However, all sectors contribute with more than $10\%$ to the observed distribution. Also for the April 2015 measurements the derived transport history is confirmed by the results from $CO_2$.

## 6 Discussion

Jiao and Flanner (2016) used the maximum zonal mean latitudinal gradient of $500\,\mathrm{hPa}$ geopotential height in the Northern Hemisphere to assess the impact of changes in atmospheric transport and removal processes due to climate change on the aerosol distribution in the Arctic. They deduced the polar dome boundary between 40 and $50\,°N$ during January which is further south compared to our tracer derived values. However, their method does not account for the lower troposphere, which is essential for the diabatic processes affecting transport into the Arctic. For January Klonecki (2003) determined a mixing barrier in the lower troposphere at $60\,°N$ at longitudes between $60\,°W$ and $105\,°W$ for a short lived artificial tracer (7 days atmospheric lifetime) emitted in North America and Europe. During summer they reported that the strong mixing barrier moves north following the location of the Arctic front. This is in the range of our horizontal polar dome boundary for the respective spring (66.0 to $68.5\,°N$) and summer ($73.5\,°N$) season. In comparison to the Arctic front our analysis seems to give a more northern boundary for July 2014 and April 2015. Furthermore, Klonecki (2003) reported an increasing mid-latitude influence with increasing altitude, in particular above $4\,\mathrm{km}$ altitude which supports our findings of the potential temperature boundary being below $300\,\mathrm{K}$ for both seasons, which corresponds to an altitude below $4\,\mathrm{km}$. The increasing mid-latitude influence with altitude is also in line with the results from Stohl (2006), who also used the Arctic front as a marker for the polar dome boundary.

The most isolated regions of the polar dome, where air masses experience the longest residence times, span a bigger area during late spring (April 2015) compared to summer (July 2014). This is in good agreement to previous studies, since it is already known that the polar dome extent is much smaller during summer compared to the winter months and processes and transport pathways controlling the composition of the high Arctic lower troposphere differ between both seasons (Klonecki, 2003; Stohl, 2006; Law and Stohl, 2007; Engvall et al., 2008; Garrett et al., 2010; Fuelberg et al., 2010). Our measurements during spring confirm a larger extent of the dome compared to summer and rather represent winter conditions than summer.

Stohl (2006) further defined an Arctic age of air and concluded that this age increased with decreasing altitude from 3 days between 5-8 km to around 1 week near the surface during the winter season (maximum 10 days in the North America region).



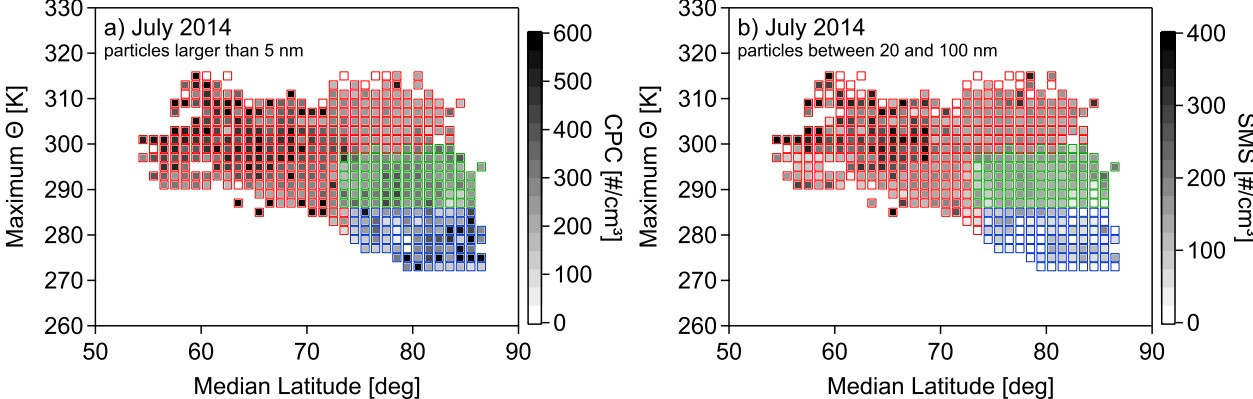

**Figure 15.** Aerosol distribution as a function of maximum potential temperature and median latitude. The median and maximum values were derived from every 10-day-trajectory calculated along the flight track. The grey-scale colour code represents the average particle concentration per bin. The coloured bin frame represent the three different regions identified in Sec. 5.3 (see Tab. 2): aged polar dome (blue), mixing region (green) and outside polar dome (red). (a) contains the distribution of aerosol with a diameter larger than 5 nm measured with a Condensation Particle Counter (CPC) and (b) contains measurements of aerosol between 20 and 100 nm obtained with a scanning mobility system (SMS) (see Burkart et al. (2017).

During the summer season the air in the lowest 100 m of the troposphere is even older with values of 13-17 days north of 75 °N. For a rough estimate of the upper limit of a transport timescale for mid-latitude air travelling into the summer polar dome during NETCARE one can estimate the time at which the average mixing ratio within the polar dome was last observed at mid-latitude ground based observatories (for example Mace Head, Ireland). For $CO_2$ this gives a transport time of around

three weeks which is in the order of magnitude of the Arctic age of air in the lowermost troposphere in the summertime Arctic reported by Stohl (2006). This assumes a transport controlled mixing ratio in the Arctic lower troposphere which is justified by studies from Fung et al. (1983), Parazoo et al. (2011) and Barnes et al. (2016). Stohl (2006) further report that the Arctic troposphere is flushed on the time scales of 1-2 weeks in winter whereas in summer the corresponding timescale is twice as long. Assuming a similar short Arctic age during the spring season this implies that above the polar dome air masses can be

transported within days from mid-latitude regions to the Arctic troposphere. Tracer concentrations will be further homogenized along isentropic surfaces when diabatic processes are slow compared to transport timescales (Klonecki, 2003). This is evident in a layer of similar CO mixing ratios above the polar dome for the April 2015 measurements (see Fig. 9c). During the July 2014 measurements increased diabatic heating due to convective and boundary layer heating in mid-latitudes can lead to an uplift of air masses at mid-latitudes and further transport into the Arctic which prevents an isentropic distribution during

summer as evident in Fig. 9a and b. Several studies analysed the transport of mid-latitude air masses into the Arctic troposphere above the polar dome along those pathways mentioned before, without specifying the extent of the polar dome in more detail (Fuelberg et al., 2010; Roiger et al., 2011; Sodemann et al., 2011; Schmale et al., 2011; Brock et al., 2011; Ancellet et al., 2014, and references therein).



Based on the 2014 data set the influence of the change in the weather regime on CO and $CO_2$ levels was quantified. Mid-latitude air masses with enhanced CO and reduced $CO_2$ concentrations most probably due to biomass burning emissions entered the high Arctic lower troposphere pushing the polar dome northward, which significantly changed the levels of the two trace gases in the measurement region (see Tab. 2). Burkart et al. (2017) also reported a significant change in the aerosol

loading during the first and the second phase of the campaign.

Having defined the polar dome based on trace gas gradients now allows for a more detailed study of aerosol within the polar dome. Efficient wet removal and less efficient transport from lower latitudes lead to generally low aerosol concentrations (Stohl, 2006; Engvall et al., 2008), especially within the Arctic lower troposphere during summer. As evident in Fig. 15a and b, the lowest aerosol concentrations (Burkart et al., 2017) were observed within the aged polar dome which was derived based

on the trace gas gradients. Interestingly Willis et al. (2017) found evidence of secondary aerosol formation events which can play an important role in growing nucleation mode particles into cloud condensation nuclei (CCN)-active sizes in the clean summertime Arctic lower atmosphere within the polar dome. These events are indicated through enhanced concentrations within the polar dome region (blue squares) in the aerosol distributions shown here. Furthermore, the aerosol data support the previously discussed threefold structure of the high Arctic lower troposphere in the Resolute Bay area north of 75 °N. This is

indicated in Fig. 15 through different concentrations levels of the aerosol particles in the respective region. Köllner et al. (2017) also observed a threefold structure in the Arctic troposphere based on the analysis of single particle measurements. The known polar dome extent is further used by Willis et al. (2019) and Schulz et al. (2018) to study vertically varying source regions and chemical processing within the polar dome.

## 7    Summary and conclusion

In this study we defined the polar dome boundary based on tracer gradients. For July 2014, the horizontal polar dome boundary was found to be at the latitude of 73.5 °N. In the vertical a threefold structure established with the strongest gradient being observed at the potential temperature range between 299 and 303.5 K separating air masses within the polar dome from those outside. A second weaker gradient was found at a potential temperature of 285 K. Below this potential temperature the region was denoted as the aged polar dome with the highest degree of isolation and thus the longest residence time of air masses

whereas above a mixing region established. The mixing region still shows significant characteristics of the polar dome region and clearly separates from outside the polar dome (see Fig. 11a). For April 2015 the tracer-derived boundary between inside the polar dome and outside was determined to be between 66.0 and 68.5 °N and in a potential temperature range of 283.5 to 287.5 K.

Using the tracer derived polar dome boundaries PDFs of CO and $CO_2$ values inside and outside the polar dome clearly

showed a difference in the distribution and also the absolute value of the distribution maximum. In the tracer-tracer scatter plot the polar dome separates from the surrounding by a different slope (April 2015) or a narrow group of data points at end of the highest $CO_2$ and lowest CO mixing ratios (July 2014). The PDFs and the scatter plots confirm the different air mass properties inside and outside of the polar dome.





The processes dominating the recent transport history were analyzed using a phase-space diagram based on Binder et al. (2017). For air masses outside the polar dome diabatic cooling and a temperature increase was prevalent in both seasons spring and summer (see Fig. 13d and Fig 14c). The associated transport pathway starts at mid-latitudes where air masses are lifted in convective or frontal systems followed by further northward motion towards the high Arctic, before the descent starts into the

lower Arctic troposphere. Predominantly North America and East Asia are identified as the source region for this pathway by previous studies (Klonecki, 2003; Stohl, 2006). These source regions for our observations are also indicated from Figs. 4 and 5, however a more comprehensive study of the source regions is beyond the scope of this paper.

For spring and summer air masses within the polar dome separate based on their transport history. During spring air masses experience predominantly diabatic cooling and lose temperature which can be associated to low level transport over cold

surfaces. During summer an efficient cooling mechanism is missing. In fact, already cold air masses within the polar dome potentially experience a weak heating thus leading to a conditionally unstable lower troposphere and potentially weak lifting. Diabatic cooling rates determined from the trajectories are in good agreement to the range of $1\,K$ per day (radiative cooling) to several degrees per day (contact with cold and mostly snow covered surface) reported for diabatic processes by Klonecki (2003).

We conclude that the different chemical composition within and outside the polar dome allows for trace gas gradient based definition of the polar dome boundary. The phase-space diagram helped to cluster the air masses based on their differing heating and cooling rates. This gives further insight in the processes that control the recent transport history of the air masses within and outside the polar dome. The polar dome boundary derived in this study is already used to study the source regions and chemical composition of aerosol within the polar dome. A polar dome boundary derived from gradients of chemical tracers can

be further used for a quantification of the influence of inner Arctic and remote sources of pollution affecting the Arctic lower troposphere in a changing climate. Additionally mixing and exchange processes along the polar dome boundary triggered by synoptic disturbances can be studied to shed light on additional pathways of mid-latitude pollution that reaches the Arctic lower troposphere.

*Data availability.* All data from NETCARE are available on the Government of Canada Open Data Portal (https://open.canada.ca/data/en/dataset).

*Author contributions.* Author contributions. HB wrote the paper, with significant conceptual input from PH and DK and critical feedback from all co-authors. HB, FK, JS, MDW, JB, HS and WRL operated instruments in the field and analyzed resulting data. AAA analyzed flight data. WRL, JPDA and ABH designed the field experiment. DK ran LAGRANTO simulations, and HB analyzed the resulting data with input from DK.

*Competing interests.* The authors declare that they have no conflict of interest.





*Acknowledgements.* The authors acknowledge a large number of people for their contributions to this work. We thank Kenn Borek Air, in particular Kevin Elke, John Bayes, Gery Murtsel and Neil Travers, for their skillful piloting that facilitated these observations. We are grateful to John Ford, David Heath and the U of Toronto machine shop, Jim Hodgson and Lake Central Air Services in Muskoka, Jim Watson (Scale Modelbuilders, Inc.), Julia Binder and Martin Gerhmann (AWI), Mike Harwood and Andrew Elford (ECCC) for their support of the

5   integration of the instrumentation and aircraft. We thank Mohammed Wasey for his support of the instrumentation during the integration and in the field. We are grateful to Carrie Taylor (ECCC), Bob Christensen (U of T), Kevin Riehl (Kenn Borek Air), Lukas Kandora, Manuel Sellmann and Jens Herrmann (AWI), Desiree Toom, Sangeeta Sharma, Dan Veber, Andrew Platt, Anne Mari Macdonald, Ralf Staebler and Maurice Watt (ECCC), Kathy Law and Jennie Thomas (LATMOS) for their support of the study. We thank the biogeochemistry department of MPIC for providing the CO instrument and Dieter Scharffe for his support during the preparation phase of the campaign. We thank the

10  Nunavut Research Institute and the Nunavut Impact Review Board for licensing the study. Logistical support in Resolute Bay was provided by the Polar Continental Shelf Project (PCSP) of Natural Resources Canada under PCSP Field Project no. 218-14, and we are particularly grateful to Tim McCagherty and Jodi MacGregor of the PCSP. Funding for this work was provided by the Natural Sciences and Engineering Research Council of Canada through the NETCARE project of the Climate Change and Atmospheric Research Program, the Alfred Wegener Institute and Environment and Climate Change Canada.



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
