# Peer review of "Characterization of Transport Regimes and the Polar Dome during Arctic Spring and Summer using in-situ Aircraft Measurements"

_Atmospheric Chemistry and Physics, 2019_

## Referee Comment (RC1) · Anonymous Referee #1 · 6 Mar 2019

This manuscript provides an analysis of air masses present during the 2014 and 2015 NETCARE airborne measurements spanning a broad region of the western Arctic between Spitsbergen and Alaska. An overview of the meteorological conditions during the summer (2014) and winter/spring (2015) campaigns is given, then trajectories are used to identify airmass history. Trace gas observations are described, then combined with potential temperature to identify regions with sharp gradients in CO and $CO2$. These gradients are used to define the "polar dome", the region of the cold, stable, near-surface Arctic airmass that is most isolated from midlatitude influences. (As this airmass wobbles over sources of pollution in the winter, it accumulates pollutants because sinks are very slow, leading to the seasonal near-surface "Arctic haze"

phenomenon, which is of broad interest.) The statistics of CO and CO2 abundance in the different regions identified from this analysis are then presented. Next, a "transport regime" analysis, based on the trajectories and using methods developed in Binder (2017), is used to evaluate the influence of lifting within and outside the Arctic, and diabatic processes, on CO and CO2 mixing ratios within and outside of the polar dome. Finally, a discussion section evaluates the abundance of nucleation and Aitken mode particles within the three regimes identified in the earlier analysis (inside the polar dome, outside the polar dome, and a mixing region).

This is an ambitious manuscript, with many parts. It has interesting sections, but it doesn't seem to have a strong overall purpose. My fundamental complaint with the manuscript is that it doesn't make the case for any generality to the analysis. Are the results more broadly applicable outside of the narrow time period and location of the NETCARE airborne observations in 2014 and 2015? For example, in Sect. 5.3 there is a long discussion of how the CO and CO2 observations can be used to identify the polar dome boundaries, and a specific range of potential temperatures and latitudes is the result of the analysis. This is great for these NETCARE observations, but are these findings more broadly applicable? For example, could one take the long-term surface observations at UtqiaÄḍvik (Barrow) or Alert or Zeppelin, apply the potential temperature and CO/CO2 screens developed in this manuscript, and separate the data out into "in the polar dome" and "out of the polar dome" datasets? This would be useful to the scientific community. Without such broader relevance, this analysis is of interest only to the very small set of scientists interested in the NETCARE data.

In addition to my concern with the applicability of the findings, I feel the manuscript also needs restructuring. The paper opens with the meteorological analysis, which is fine. Next, though, is the trajectory analysis. It would be more logical if the next section were the identification of the polar dome using the trace gas gradients. Then the backtrajectories could follow, with backtrajectories initiated either within the polar dome, outside of it, or in the mixing region. The trajectories would then provide an independent and

intuitive confirmation of the identification of the polar dome that was derived from the trace gas and potential temperature data. Section 5.4, which is a presentation of PDFs of CO and CO2 from the three different airmasses is not very logical. CO and CO2 were used to identify the three airmass classifications, after all, so it's entirely expected that they would have different PDFs–the reasoning is circular. Instead of this section, the next logical section would be examining the transport regimes using the Binder et al. methodology, as it continues the analysis of trajectories. This section could be made stronger by coupling the delta-theta/delta-T plots with graphs of trajectory clusters (e.g., plots trajectory altitudes as a function of latitude). This would bolster the rather speculative discussion about the meaning of each of the sectors of the Binder plots. This section could get rather long, so it might make sense to give one example (from the springtime flights, perhaps) and place the rest in the supporting materials. This analysis is the part of the current manuscript that is really informative outside of the narrow range of the NETCARE project, as it suggest broader generalities about how transport occurs into the polar dome. I'd like to see it developed more thoroughly and the conclusions made firmer, perhaps with a concluding paragraph that summarizes the findings from this section.

Following the transport regime analysis, the next logical section would be to see how these different airmass types are manifested in the pollution loadings. This is currently Sect. 6, which is labelled "discussion". I appreciate the novelty of Fig. 15, but it's very difficult to understand the grey-scale coloring on top of the colored classification scheme. I'd much rather see PDFs of the number concentrations in the 5-20 and 20-100 and >100 size class. Are there also observations of aerosol composition (e.g., BC abundance, composition) that could be added to this section? You've effectively classified the measurements into airmass type; it would be extremely interesting to see how all the available aerosol microphysical and chemical parameters vary within these different airmasses, and compare them with existing literature values.

Finally, if it's feasible it would be wonderful to extend this analysis to the surface data

from the long-term monitoring sites. This would show that the classifications developed here are more broadly applicable. At least an evaluation of whether the approach here is applicable to other cases, or is specific to NETCARE, is needed.

In addition to these larger, structural issues, the manuscript needs some technical correction. The primary author is not a native English speaker, but English-speaking co-authors should step up and give the manuscript a round of thorough copy-editing. Verb tense is not used consistently, which is distracting and sometimes confusing. (Example, p. 17 lines 18-21 go from "now calculated" to "determine" to "finally used".) There are quite a few typos that a spell checker should find, and terms like "surface-near" instead of "near-surface" are present.

This manuscript has a lot of good analysis using an interesting and unique dataset. The CO and CO2 measurements look spot-on with the long-term ground monitoring network data, which is very encouraging since these airborne measurements can be tricky given the large background. With the suggested restructuring and a tighter focus on how the findings are more broadly applicable, the manuscript should be quite suitable to publish in ACP.

---

## Referee Comment (RC2) · Anonymous Referee #2 · 16 Mar 2019

This work makes use of the atmospheric tracers CO and CO2 measured on two NET-CARE flight campaigns together with 10-day back trajectories to describe air mass transport into the high Arctic during spring and summer. The authors find distinct transition zones between the mid-latitudes and the polar dome for the two seasons based on tracer gradients. The tracer derived polar dome boundaries are subsequently applied to aerosol number concentration data. In addition, the authors explore different transport pathways of air masses into the Arctic using a previously published phase-space approach that relates the maximum change in potential and absolute temperature along the trajectory.

[Figure]

This manuscript is very comprehensive and represents a novel approach to localize the polar dome boundary and to characterize air mass transport into the Arctic. The work shall definitely be published with revisions as described below.

General remarks

To give the findings more relevance for Arctic atmospheric research it would be of high interest, whether the derived polar dome boundaries are representative for other Arctic sectors and years as well? Or would each field campaign have to do their own analysis following this example? So a discussion on how far the results can be generalized is needed.

There is no discussion on the uncertainties of the variables along the trajectory such as potential and absolute temperature. Particularly the vertical location of the polar dome boundary would be subject to the uncertainty. At least some discussion on how the ECMWF input data compares to the in-situ measurements should be added.

The application of the polar dome boundary to aerosol data is intriguing and it would be very interesting to know how other aerosol properties relate to the boundary. Adding such information to the manuscript would make it even longer than it already is. I would recommend exploring whether a "part 2" manuscript on aerosols could make sense.

The manuscript is partly repetitive and the abstract reads almost like an introduction. Both the abstract and the whole manuscript should be shortened. There are some recommendations in the attached PDF.

Specific comments p. 7, l. 15: What is meant by "The stability of the instrument. . ."? Is this the accuracy? I wonder whether none of the instruments has been described before and whether such an extensive description here is necessary?

p. 8, l. 19: What is a "very stable stratification" compared to a stable stratification? Is the ECMWF data the analysis or re-analysis data?

Figure 6: The scales can be enlarged in both panels: 200 for a) and 415 for b). This way, more details would be visible. The legends will find place somewhere else. ... What is the purpose of the figure except discussing the seasonal cycle? If a direct comparison of the NETCARE data with the stations is the goal, there should be zooms for the short periods of time. Currently, one cannot see much because the symbols are so large and cover everything. Why is Mace Head chosen as a reference?

Figure 8 b: Why does the potential temperature increase below which CO maximum concentrations occur with decreasing latitude? Some explanation is needed.

p. 28, l. 6-18: This paragraph is a result and should be moved to the results section instead of being added to the discussions.

p. 28, l. 10-13: How can you infer that secondary aerosol formation is responsible for the concentration difference in the blue area? Based on the information provided, those particles are either between 5 and 20 nm or > 100 nm. For the first option, I find it difficult to believe that there is evidence to relate the increase in 5 to 20 nm particles to secondary aerosol formation based on the AMS and ALABAMA measurements in Willis et al. (2017). These instruments do not cover the relevant size range. If the second option is true, the information on SOA contributing to particle growth to form CCN does not make sense, because particles are already in the CCN size range, even for low supersaturations. What is the explanation for the difference between panel a and b? And what is the relevance for CCN? Please revise the statement.

Figure 15: The way the aerosol results are presented with the colored boxes makes it difficult to see the shading.

Technical comments:

Please the comments in the attached PDF manuscript.

Please also note the supplement to this comment:

https://www.atmos-chem-phys-discuss.net/acp-2019-70/acp-2019-70-RC2-supplement.pdf

[Figure]

**Supplement:**

[revised manuscript text omitted]

---

## Author Comment (AC1) · 8 Aug 2019

This manuscript provides an analysis of air masses present during the 2014 and 2015 NETCARE airborne measurements spanning a broad region of the western Arctic between Spitsbergen and Alaska. An overview of the meteorological conditions during the summer (2014) and winter/spring (2015) campaigns is given, then trajectories are used to identify airmass history. Trace gas observations are described, then combined with potential temperature to identify regions with sharp gradients in CO and $CO_2$. These gradients are used to define the "polar dome", the region of the cold, stable, near-surface Arctic airmass that is most isolated from midlatitude influences. (As this airmass wobbles over sources of pollution in the winter, it accumulates pollutants because sinks are very slow, leading to the seasonal near-surface "Arctic haze" phenomenon, which is of broad interest.) The statistics of CO and $CO_2$ abundance in the different regions identified from this analysis are then presented. Next, a "transport regime" analysis, based on the trajectories and using methods developed in Binder (2017), is used to evaluate the influence of lifting within and outside the Arctic, and diabatic processes, on CO and $CO_2$ mixing ratios within and outside of the polar dome. Finally, a discussion section evaluates the abundance of nucleation and Aitken mode particles within the three regimes identified in the earlier analysis (inside the polar dome, outside the polar dome, and a mixing region).

This is an ambitious manuscript, with many parts. It has interesting sections, but it doesn't seem to have a strong overall purpose. My fundamental complaint with the manuscript is that it doesn't make the case for any generality to the analysis. Are the results more broadly applicable outside of the narrow time period and location of the NETCARE airborne observations in 2014 and 2015? For example, in Sect. 5.3 there is a long discussion of how the CO and $CO_2$ observations can be used to identify the polar dome boundaries, and a specific range of potential temperatures and latitudes is the result of the analysis. This is great for these NETCARE observations, but are these findings more broadly applicable? For example, could one take the long-term surface observations at UtqiaÄa˛vik (Barrow) or Alert or Zeppelin, apply the potential temperature and $CO/CO_2$ screens developed in this manuscript, and separate the data out into "in the polar dome" and "out of the polar dome" datasets? This would be useful to the scientific community. Without such broader relevance, this analysis is of interest only to the very small set of scientists interested in the NETCARE data.

We appreciate the reviewer's comments and the valuable suggestions for restructuring the manuscript which we largely followed and extended the discussion of the transport regimes. Regarding the point of a larger scale applicability of the method we included a first comparison with station data from Zeppelin in the reply.

In this particular paper our intension was indeed to develop an empirical dome boundary based on the airborne measurements of tracer like CO and $CO_2$ for NETCARE and to show first applications. We will extend the method and apply it to other airborne data sets or ground based data. Note however, that ground based data could eventually have a different tracer characteristics since these data are stronger affected by local sources and sinks and partly decoupled from the free troposphere by the boundary layer inversion. Nonetheless, we tested the approach for Zeppelin, which provides very promising results (see below). We will publish a follow-up paper, which will apply the method to a larger data set to investigate the general applicability.

In addition to my concern with the applicability of the findings, I feel the manuscript also needs restructuring. The paper opens with the meteorological analysis, which is fine. Next, though, is the trajectory analysis. It would be more logical if the next section were the identification of the polar dome using the trace gas gradients. Then the backtrajectories could follow, with backtrajectories initiated either within the polar dome, outside of it, or in the mixing region. The trajectories would then provide an independent and intuitive confirmation of the identification of the polar dome that was derived from the trace gas and potential temperature data.

We restructured the manuscript mostly as suggested: We moved the detailed trajectory analysis of former Figure 4 and 5 to the supplement and combined the chemical regime discussion based on trace gases with the (newly restructured) aerosol paragraph at the end. This results in a structure as suggested: Meteorological analysis, determination of the polar dome boundary, transport regimes and air mass history, chemical regimes. Note that Willis et al. (2019) and Schulz et al. (2019) already used the polar dome boundary presented in this study to analyse aerosol composition, transport and sources within the springtime polar dome and within and partly outside the summertime polar dome, respectively. For a more detailed aerosol analysis we refer to their studies and only briefly address characteristics of aerosol within this study.

Section 5.4, which is a presentation of PDFs of CO and $CO_2$ from the three different air masses is not very logical. CO and $CO_2$ were used to identify the three air mass classifications, after all, so it's entirely expected that they would have different PDFs–the reasoning is circular.

It is important to keep in mind that the dome boundaries were determined based on the gradients of CO and $CO_2$, not applying any threshold value for tracers. In the case of horizontal (i.e. isentropic) gradients the analysis was performed for different isentropes in intervals of 2K, allowing for different thresholds for each isentropic interval. There is no step within the analysis, which makes use of an absolute value of either CO or $CO_2$. Since the gradient is not related to an absolute value of CO or $CO_2$, trace gas values and particular the variability therefore still provide additional information for the regimes. The width of the distributions should be larger outside the dome regions, the mean (median values) resemble the surface network data. We therefore kept this section since it illustrates the chemical properties of the regimes. As mentioned before it is moved to the end of the manuscript.

Instead of this section, the next logical section would be examining the transport regimes using the Binder et al. methodology, as it continues the analysis of trajectories.

After restructuring the paper, the next section that follows is the analysis of the transport regime based on the Binder et al. methodology as suggested by the reviewer.

This section could be made stronger by coupling the delta-theta/delta-T plots with graphs of trajectory clusters (e.g., plots trajectory altitudes as a function of latitude). This would bolster the rather speculative discussion about the meaning of each of the sectors of the Binder plots. This section could get rather long, so it might make sense to give one example (from the springtime flights, perhaps) and place the rest in the supporting materials. This analysis is the part of the current manuscript that is really informative outside of the narrow range of the NETCARE project, as it suggest broader generalities about how transport occurs into the polar dome. I'd like to see it

developed more thoroughly and the conclusions made firmer, perhaps with a concluding paragraph that summarizes the findings from this section.

We included the figures according to the suggestion showing the evolution of the trajectories. As an example, sector 1 of the April 2015 measurements was chosen for those data points inside the polar dome, since it is the most dominant sector. In contrast, sector 4 was chosen for the air masses outside the dome, representing the largest contribution there. For better figure clarity only every 20[th] individual trajectory is plotted. Corresponding figures for the dominating sectors for July 2014 are included in the supplement material.

New figures for trajectory analysis inside the polar dome. Shown is the dominating sector 1 from Fig. 11b in the paper:

[Figure]

*Figure 12: (a) Trajectories of the most dominant sector 1 for air masses inside the polar dome. The color code represents the pressure along the trajectories. (b) The same trajectories as in (a) as a function of pressure and latitude, color coded by potential temperature. In both figures (a) and (b) black circles denote the initialisation point of the trajectory along the flight track. The black open squares show the position of the trajectory 10 days back in time. Figures (c) and (d) show a vertical cross section of the trajectory evolution over the 10 days of travel with the color code denoting the temperature (c) and potential temperature (d). The black line marks the median pressure of the trajectory cluster at the individual time steps and the grey line indicates the median temperature and median potential temperature, respectively. Note that in all figures only every 20th trajectory is plotted for figure clarity.*

New figures for trajectory analysis outside the polar dome showing the dominating sector 4 from Fig. 11c in the paper:

[Figure]

*Figure 13: (a) Trajectories of the most dominant sector 4 for air masses outside the polar dome. The color code represents the pressure along the trajectories. (b) The same trajectories as in (a) as a function of pressure and latitude, color coded by potential temperature. In both figures (a) and (b) black circles denote the initialisation point of the trajectory along the flight track. The black open squares show the position of the trajectory 10 days back in time. Figures (c) and (d) show a vertical cross section of the trajectory evolution over the 10 days of travel with the color code denoting the temperature (c) and potential temperature (d). The black line marks the median pressure of the trajectory cluster at the individual time steps and the grey line indicates the median temperature and median potential temperature, respectively. Note that in all figures only every 20th trajectory is plotted for figure clarity.*

The following paragraph was added to the manuscript:
*Based on the results from the phase space diagrams we further analyze the trajectories of the individual clusters. This allows for a more detailed analysis of the physical processes along the trajectory. Therefore, we compare the two most dominant sectors for the April 2015 measurements in Figs. 12a-d and 13a-d. Sector 1 is mostly dominated by air masses confined to the central Arctic at all altitude levels (see Figs. 12a and b). The air masses show a weak descent during the 10 days before the measurements but experience a very pronounced decrease in temperature and potential temperature indicated by the evolution of median temperature of the whole trajectory cluster (see Figs. 12c and d). In contrast, the trajectory analysis of the most dominant sector 4 of air masses*

*outside the polar dome reveals different air streams contributing to the cluster. Air masses originate at different altitudes in the central Arctic, at low level over the Pacific Ocean and from the upper troposphere over Asia. Air masses in this cluster are characterized by a significant increase of median temperature and decrease of median potential temperature indicating a descending trend, which is confirmed by the median pressure decrease over the time of travel. The low-level transport over the Pacific is associated with a low-pressure system over Alaska. Those air masses arrive at the polar dome boundary in the measurement region after experiencing a week net cooling over Alaska.*

*We conclude that air masses within the aged summertime polar dome are mostly confined to the boundary layer while they experienced a week diabatic warming due to insolation in July 2014 during NETCARE. In the mixing region and outside the polar dome diabatic cooling and a continuous descent is observed. Within the polar dome in April 2015 during NETCARE mostly near-surface processes (diabatic cooling due to the flow over cold surfaces) dominate the recent transport history of air masses in the lower polar dome. Air masses in the upper polar dome experience a very slow descent induced by radiative cooling. Outside the polar dome air masses mostly arrive at higher potential temperatures in the Arctic and experience a continuous slow descent with increasing temperatures but only week diabatic cooling.*

Following the transport regime analysis, the next logical section would be to see how these different airmass types are manifested in the pollution loadings. This is currently Sect. 6, which is labelled "discussion". I appreciate the novelty of Fig. 15, but it's very difficult to understand the grey-scale coloring on top of the colored classification scheme. I'd much rather see PDFs of the number concentrations in the 5-20 and 20-100 and >100 size class. Are there also observations of aerosol composition (e.g., BC abundance, composition) that could be added to this section? You've effectively classified the measurements into airmass type; it would be extremely interesting to see how all the available aerosol microphysical and chemical parameters vary within these different airmasses, and compare them with existing literature values.

We thank the reviewer for this point and revised the paragraph in the revised manuscript. We just want to show the relevance of our empirical dome boundary to other applications using aerosol formation as an example. The specific analysis and discussion of aerosol processes is beyond the scope of this paper. We refer to Willis et al. (2019) and Schulz et al. (2019) for a more detailed discussion on aerosol composition within and outside the polar dome. They used the polar dome boundary derived in this study for their analysis. We revised the paragraph on aerosol as follows including a new figure 15:

[Figure]

[Figure]

[Figure]

*Figure 15: Normalized probability density functions for particles with diameters between 5 and 20 nm (a) and larger than 100 nm for July 2014 (b). The colour code represents the three different regions identified during the polar dome analysis. Panel (c) shows the distribution of the fraction of particles containing trimethylamine (TMA) with respect to the total amount of particles measured by single particle mass spectrometry (Köllner et al., 2017) in the maximum potential temperature and median latitude coordinate system. Note that for the reason of particle spectra statistics the resolution of median latitude and maximum potential temperature interval is reduced compared to the refractory black carbon distribution (BC) shown in panel (d). The enhancement in BC within the polar dome is most probably due to fresh local pollution.*

*Having defined the polar dome based on trace gas gradients now allows for a more detailed study on aerosol associated with the different air masses. Efficient wet removal and less efficient transport from lower latitudes lead to generally low aerosol concentrations (Stohl, 2006; Engvall et al., 2008), especially within the Arctic lower troposphere during summer. This is consistent with results in Fig. 15a and b. Observations of elevated levels of accumulation mode particles ($N_{>100}$) can be associated with regions outside the polar dome and subsequent transport to the measurement region. In parallel, regions within the polar dome were characterized by $N_{>100}$ smaller than 100 $cm^{-3}$. In contrary, number concentrations of ultrafine particles ($N_{5-20}$) showed occasionally larger values within the polar dome compared to outside (Fig. 15a), indicating the formation of ultrafine particles occurred within the polar dome region (Burkart et al., 2017). Exemplary for aerosol composition, particulate trimethylamine (measured by single particle mass spectrometry) can be associated with sources within the polar dome (Fig. 15c), consistent with results in Köllner et al. (2017) and Willis et al. (2017). In contrast, the abundance of refractory black carbon can be linked to pollution sources outside the polar dome and subsequent transport to the measurement region (Fig. 15d; Schulz et al., 2019). To conclude, the method introduced in this study is a useful tool to combine Arctic aerosol observations with transport processes and sources within and outside the polar dome region.*

Finally, if it's feasible it would be wonderful to extend this analysis to the surface data from the long-term monitoring sites. This would show that the classifications developed here are more broadly applicable. At least an evaluation of whether the approach here is applicable to other cases, or is specific to NETCARE, is needed.

In a follow-up paper to this study we will extend our approach to a variety of different field campaigns in the Arctic covering different seasons and different locations. To demonstrate the applicability of the tracer based diagnostics to a broader data set we analyzed two examples of hourly ground based observations at the Zeppelin Mountain observatory (Ny Alesund, Spitsbergen) (see Fig. R2). Based on $CO$-$CO_2$-scatter plots the signatures and characteristics of both species inside the polar dome during the campaign phases are also found in the respective ground based observations. Note that ground based and airborne observations are in principle affected by different

processes and not necessarily linked. Particularly, the CO and $CO_2$ data at higher potential temperatures during the airborne campaigns are linked to higher altitudes as evident in both Figs. R1 a and b.

[Figure]

Figure R1 (a): Tracer-tracer scatter plot of all aircraft data points (background + pollution plumes) for July 2014. (b): Tracer-tracer scatter plot of all aircraft data points (background + pollution plumes) for April 2015. The color code denotes potential temperature.

[Figure]

Figure R2 (a): Tracer-tracer scatter plot of all aircraft data points (background + pollution plumes) within the aged polar dome (blue), the mixing region (green) and outside the polar dome (red) for July 2014. (b): Tracer-tracer scatter plot of all aircraft data points (background + pollution plumes) within (blue) and outside (red) the polar dome for April 2015. To separate the different regions the tracer derived polar dome boundaries are used. Boundary values for each region are summarized in Tab. 2 in the paper. The black circles denote the ground based observation data from the Zeppelin mountain observatory (Ny Alesund, Spitzbergen) for the months July (a) and April (b) 2014 and 2015.

Note however, the good agreement at low potential temperatures particularly for the April 2015 case with exactly the same slope and similar absolute values at the station as deduced from the aircraft data as inside dome (see Figs R2 a and b). Based on this analysis we conclude that the Zeppelin observatory was inside the polar dome for April 2014 and 2015 and most probably inside the polar dome with episodes outside the polar dome in July 2014 and 2015. The latter is indicated by the correlation following the characteristics of the mixing region (see Fig. R2a). However, potential temperatures were as low as those observed during the research flights (not shown).

In addition to these larger, structural issues, the manuscript needs some technical correction. The primary author is not a native English speaker, but English-speaking co-authors should step up and give the manuscript a round of thorough copy-editing. Verb tense is not used consistently, which is distracting and sometimes confusing. (Example, p. 17 lines 18-21 go from "now calculated" to "determine" to "finally used".) There are quite a few typos that a spell checker should find, and terms like "surfacenear" instead of "near-surface" are present. This manuscript has a lot of good analysis using an interesting and unique dataset. The CO and CO2 measurements look spot-on with the long-term ground monitoring network data, which is very encouraging since these airborne measurements can be tricky given the large background. With the suggested restructuring and a tighter focus on how the findings are more broadly applicable, the manuscript should be quite suitable to publish in ACP.

We thank the reviewer for this point. Copy-editing for the manuscript will be done by native English speakers.

Acknowledgement:

All atmospheric data from Zeppelin are publicly available in the EBAS database (http: //ebas.nilu.no) and we thank Cathrine Lund Myhre and NILU - Norwegian Institute for Air Research for making the CO and $CO_2$ observations from Zeppelin available.

---

## Author Comment (AC2) · 8 Aug 2019

This work makes use of the atmospheric tracers CO and CO2 measured on two NETCARE flight campaigns together with 10-day back trajectories to describe air mass transport into the high Arctic during spring and summer. The authors find distinct transition zones between the mid-latitudes and the polar dome for the two seasons based on tracer gradients. The tracer derived polar dome boundaries are subsequently applied to aerosol number concentration data. In addition, the authors explore different transport pathways of air masses into the Arctic using a previously published phasespace approach that relates the maximum change in potential and absolute temperature along the trajectory.

This manuscript is very comprehensive and represents a novel approach to localize the polar dome boundary and to characterize air mass transport into the Arctic. The work shall definitely be published with revisions as described below.

General remarks:

To give the findings more relevance for Arctic atmospheric research it would be of high interest, whether the derived polar dome boundaries are representative for other Arctic sectors and years as well? Or would each field campaign have to do their own analysis following this example? So a discussion on how far the results can be generalized is needed.

This study was intended to derive a tracer based diagnostic for the determination of the polar dome boundary. In a follow-up paper to this study we will discuss the application of the metrics to a comprehensive data set consisting of several field campaigns in the Arctic covering different seasons and different locations.
To demonstrate the general applicability of our campaign-based approach we analyzed two years of hourly ground based observations at the Zeppelin Mountain observatory (Ny Alesund, Spitsbergen) to show a more generalization of the tracer-based diagnostic (see Figs. R2 a and b). According to our campaign-based findings using the CO-$CO_2$ relationship the analysis of the ground-based data confirm the signatures and characteristics of both species inside the polar dome during the campaign. This potentially allows for a determination of the station location relative to the polar dome. Note, however, that ground based observations are in principle strongly affected by local sources and sinks. Depending on e.g. the strength of the boundary layer inversion these effects decouple the free troposphere above from these observations within the planetary boundary layer below.

[Figure]

Figure R1 (a): Tracer-tracer scatter plot of all aircraft data points (background + pollution plumes) for July 2014. (b): Tracer-tracer scatter plot of all aircraft data points (background + pollution plumes) for April 2015. The color code denotes the potential temperature.

[Figure]

Figure R2 (a): Tracer-tracer scatter plot of all aircraft data points (background + pollution plumes) within the aged polar dome (blue), the mixing region (green) and outside the polar dome (red) for July 2014. (b): Tracer-tracer scatter plot of all aircraft data points (background + pollution plumes) within (blue) and outside (red) the polar dome for April 2015. To separate the different regions the tracer derived polar dome boundaries are used. Boundary values for each region are summarized in Tab. 2 in the paper. The black circles denote the ground based data from the Zeppelin mountain observatory (Ny Alesund, Spitzbergen) for the months July (a) and April (b) 2014 and 2015.

Note however, the good agreement at low potential temperatures particularly for the April 2015 case with exactly the same slope and similar absolute values at the station as deduced from the aircraft data as inside dome (see Figs R2a and b). Based on this analysis we conclude that the Zeppelin observatory was inside the polar dome for April 2014 and 2015 and most probably inside the polar dome with episodes outside the polar dome in July 2014 and 2015. The latter is indicated by the correlation following the characteristics of the mixing region (see Fig. R2a). However, potential temperatures were as low as those observed during the research flights (not shown).

There is no discussion on the uncertainties of the variables along the trajectory such as potential and absolute temperature. Particularly the vertical location of the polar dome boundary would be subject to the uncertainty. At least some discussion on how the ECMWF input data compares to the in-situ measurements should be added.

The figures below show a comparison of temperature based on in-situ data and temperature derived from the analysis data interpolated to the time and location of the measurement. Lagranto backward trajectories were initialised every 10s along the flight track based on GPS horizontal coordinates and pressure. Therefore, the variability of potential temperature is driven by the temperature difference between observation and analysis. To quantify this temperature difference, we analysed the temperature for July 2014 and April 2015. For the measurements in July 2014 the median difference between in-situ temperature data and analysis data is 0.31 °C (interquartile range: -0.71 - 1.72°C). For the April 2015 measurements, the respective median difference is 1.50°C (interquartile range: 0.69 - 2.14 °C). Thus, we conclude, that the local variability of the dome boundary is largely driven by the variability of tracer gradients and the uncertainty of transport history backward in time. The latter, however, we try to evaluate (at least qualitatively) by using median latitude and maximum potential temperature. Linking these to our trace gas observations should in turn resemble the latitudinal and vertical gradient and thus account for transport history.

[Figure]

Figure R3: (a): Scatter plot of in-situ ambient temperature measured on the aircraft and temperature from the analysis data set at the time of the initialisation of the trajectories along the flight track for July 2014. (b): Probability density function of the difference between in-situ and analysis temperature data set. (c): Box and whisker plot for the difference between in-situ and analysis temperature data set.

[Figure]

Figure R4: (a): Scatter plot of in-situ ambient temperature measured on the aircraft and temperature from the analysis data set at the time of the initialisation of the trajectories along the flight track for April 2015. (b): Probability density function of the difference between in-situ and analysis temperature data set. (c): Box and whisker plot for the difference between in-situ and analysis temperature data set.

We added the following to paragraph 3.3:

*As a measure for the uncertainty of the temperature along the trajectory we calculated the median difference between temperatures measured in-situ on the aircraft and the corresponding temperatures interpolated to the initialisation point of the trajectory along the flight track based on analysis data. For the measurements in July 2014 the median is 0.31 °C (interquartile range: -0.71 - 1.72°C). For the April 2015 measurements, the respective median difference is 1.50°C (interquartile range: 0.69 - 2.14 °C)*

The application of the polar dome boundary to aerosol data is intriguing and it would be very interesting to know how other aerosol properties relate to the boundary. Adding such information to the manuscript would make it even longer than it already is. I would recommend exploring whether a "part 2" manuscript on aerosols could make sense. The manuscript is partly repetitive and the abstract reads almost like an introduction. Both the abstract and the whole manuscript should be shortened. There are some recommendations in the attached PDF.
We thank the reviewer for this point and refer to a later comment.

Specific comments:

p. 7, l. 15: What is meant by "The stability of the instrument. . ."? Is this the accuracy? I wonder whether none of the instruments has been described before and whether such an extensive description here is necessary?
The instrument is regularly calibrated during the flights to check for longer term drifts. We use these drifts as a measure for the reproducibility, which we term stability. Note that post processing accounts for these slow drifts assuming a linear drift between the calibrations. Accuracy is further affected by the uncertainty of the calibration standards. For clarification, we added the following sentence:

*"Stability is a measure for reproducibility and based on the mean drift between two subsequent calibrations which were performed during flights".*

The instrument and measurement principle of the Aerolaser ultra-fast carbon monoxide (CO) monitor model AL 5002 were described extensively in for example two papers by Gerbig et al. (1999) and Scharffe et al. (2013). The Licor LI-7200 closed $CO_2/H_2O$ analyzer from LI-COR Biosciences GmbH, modified for airborne use, was only briefly described in earlier studies in the Arctic in the framework of the NETCARE project. We follow the suggestion of the reviewer and shorten the description of the carbon monoxide monitor only leaving the specification of uncertainties for this study. We kept the more extensive description of the $CO_2$ monitor since it is the first time of a more comprehensive description of the current setup.

p. 8, l. 19: What is a "very stable stratification" compared to a stable stratification? Is the ECMWF data the analysis or re-analysis data?
The term "stable stratification" is appropriate and the text is changed accordingly. ECMWF data is operational analysis data.

Figure 6: The scales can be enlarged in both panels: 200 for a) and 415 for b). This way, more details would be visible. The legends will find place somewhere else… What is the purpose of the figure except discussing the seasonal cycle? If a direct comparison of the NETCARE data with the stations is the goal, there should be zooms for the short periods of time. Currently, one cannot see much because the symbols are so large and cover everything. Why is Mace Head chosen as a reference?
The purpose of the figure is to show both the seasonal cycle of both species CO and $CO_2$ in the Arctic as well as in the mid-latitudes and to illustrate the latitudinal gradient between the Arctic stations and mid-latitude stations. For figure clarity, only Mace Head is shown representing a mid-latitude station. It was chosen because it is a GAW station with CO and $CO_2$ data available at the respective

time period and Mace Head is furthermore located at one of the entry routes of mid-latitude air masses that frequently enter the Arctic. The comparison between ground based observations and airborne data is not the prime purpose of the figure. Therefore, we show a more extended time period and not only a zoom on the period of the measurement campaign. Figures 6 (a) and (b) were modified for figure clarity and are now Figs 4 (a) and (b) after restructuring the manuscript.

[Figure]

*Figure 4. CO (a) and CO₂ (b) seasonal cycle based on NOAA ground based measurements in Alert (Canada), Barrow (Alaska) and Mace Head (Ireland) for the years 2014 and 2015. Running means are shown for the respective station data (symbols). Mean aircraft data for altitudes < 200 m for individual flights are overlaid. Error bars (yellow and orange shading) for the aircraft data are too small to be visible. NETCARE 2014 data are in yellow and NETCARE 2015 data are in orange.*

Figure 8 b: Why does the potential temperature increase below which CO maximum concentrations occur with decreasing latitude? Some explanation is needed.

The measurements at lower latitudes were all performed from Inuvik. At that time generally higher temperatures were observed in Inuvik compared to the Arctic at similar altitude levels. Furthermore, the CO maxima in the two respective regions have different causes. In the Arctic lower troposphere high CO values are observed at low potential temperatures inside the polar dome as remnants of the wintertime maximum of CO due to the weak photochemical activity. In contrast enhanced CO values at lower latitudes are mainly caused by recent emissions (1-2 weeks old) potentially transported from Asian source regions into the measurement region by long-range transport. Since potential temperature at the surface increases in general with decreasing latitude (due to increasing temperature, see Fig. 5 in the paper) the CO maxima originating from lower latitudes affect higher isentropes. In addition, warm conveyor belt (WCB) type transport may have occurred leading to diabatic transport of CO to even higher altitudes and higher potential temperatures

p. 28, l. 6-18: This paragraph is a result and should be moved to the results section instead of being added to the discussions.

The paragraph on aerosol was located in the discussion section since it was intended to illustrate the relation of our empirically derived dome boundary when investigating e.g. aerosol processes. We followed the reviewer's suggestion and moved the paragraph to the results section and revised the whole paragraph including figures.

p. 28, l. 10-13: How can you infer that secondary aerosol formation is responsible for the concentration difference in the blue area? Based on the information provided, those particles are either between 5 and 20 nm or > 100 nm. For the first option, I find it difficult to believe that there is evidence to relate the increase in 5 to 20 nm particles to secondary aerosol formation based on the AMS and ALABAMA measurements in Willis et al. (2017). These instruments do not cover the

relevant size range. If the second option is true, the information on SOA contributing to particle growth to form CCN does not make sense, because particles are already in the CCN size range, even for low supersaturations. What is the explanation for the difference between panel a and b? And what is the relevance for CCN? Please revise the statement.

We thank the reviewer for this point and revised the paragraph in the revised manuscript. We just want to show the relevance of our empirical dome boundary for other applications using aerosol formation as an example. The specific analysis and discussion of aerosol processes is beyond the scope of this paper. We modified the paragraph as follows:

[Figure]

*Figure 15: Normalized probability density functions for particles with diameters between 5 and 20 nm (a) and larger than 100 nm for July 2014 (b). The colour code represents the three different regions identified during the polar dome analysis. Panel (c) shows the distribution of the fraction of particles containing trimethylamine (TMA) with respect to the total amount of particles measured by single particle mass spectrometry (Köllner et al., 2017) in the maximum potential temperature and median latitude coordinate system. Note that for the reason of particle spectra statistics the resolution of median latitude and maximum potential temperature interval is reduced compared to the refractory black carbon distribution (BC) shown in panel (d). The enhancement in BC within the polar dome is most probably due to fresh local pollution.*

*Having defined the polar dome based on trace gas gradients now allows for a more detailed study on aerosol associated with the different air masses. Efficient wet removal and less efficient transport from lower latitudes lead to generally low aerosol concentrations (Stohl, 2006; Engvall et al., 2008), especially within the Arctic lower troposphere during summer. This is consistent with results in Fig. 15a and b. Observations of elevated levels of accumulation mode particles ($N_{>100}$) can be associated*

*with regions outside the polar dome and subsequent transport to the measurement region. In parallel, regions within the polar dome were characterized by $N_{>100}$ smaller than 100 cm$^{-3}$. In contrary, number concentrations of ultrafine particles ($N_{5-20}$) showed occasionally larger values within the polar dome compared to outside (Fig. 15a), indicating the formation of ultrafine particles occurred within the polar dome region (Burkart et al., 2017). Exemplary for aerosol composition, particulate trimethylamine (measured by single particle mass spectrometry) can be associated with sources within the polar dome (Fig. 15c), consistent with results in Köllner et al. (2017) and Willis et al. (2017). In contrast, the abundance of refractory black carbon can be linked to pollution sources outside the polar dome and subsequent transport to the measurement region (Fig. 15d; Schulz et al., 2019). To conclude, the method introduced in this study is a useful tool to combine Arctic aerosol observations with transport processes and sources within and outside the polar dome region.*

Figure 15: The way the aerosol results are presented with the colored boxes makes it difficult to see the shading.
Figure 15 is replaced (see new figures above) to demonstrate the applicability of our empirical dome boundary to other constituents or processes using the aerosol formation just as an example.

Technical comments:

Please the comments in the attached PDF manuscript.

Please also note the supplement to this comment:

We thank the reviewer for the technical comments, which were implemented!

Acknowledgement:

All atmospheric data from Zeppelin are publicly available in the EBAS database (http: //ebas.nilu.no) and we thank Cathrine Lund Myhre and NILU - Norwegian Institute for Air Research for making the CO and $CO_2$ observations from Zeppelin available.

---

## Author Response (AR2)

We thank the reviewer for his/her comments. The comments by the reviewer are given below in black, our responses are marked in red and the corresponding changes to the manuscript are in *italics*.

**Review of "Characterization of transport regimes and the polar dome during Arctic spring and summer using in-situ aircraft measurements" by H. Bozem et al.**

The revised manuscript is much improved and the structure allows for a more logical flow. Thanks very much for undertaking the substantial task of reorganization.

The manuscript now places the analysis in a somewhat broader context, so the scope should be more relevant to more readers. With corrections and responses to minor comments as indicated below, it should be suitable for publication in ACP. I have two more substantial remaining concerns. The first is the need to clarify some terms that are not well defined--the "polar dome" vs "Arctic front". These are used somewhat interchangeably. Are they really the same thing, and if not, how do they differ? This needs a paragraph to explain their relationship. The second is related; it is the concept of springtime transport of pollution "into the Arctic airmass" from midlatitude sources. My understanding is that the Arctic front meanders southward and emissions from high midlatitude sources can directly enter the Arctic airmass--and presumably within the "polar dome" as well. This is not really transport "to" the Arctic. It would be good to see these two topics, which are closely related, explicitly adressed in a couple of short paragraphs in the Introduction. Then these concepts should be discussed with consistent terminology throughout the text.

1) p.1,line 3, change "transport barrier" to "barrier to horizontal transport"
2) p. 1, line 6, hyphenate "aircraft-based"
3) p. 1, line 11, hyphenate "gas-based"
4) p. 2, line 2, hyphenate "amine-containing"
5) p. 2, line 2, change to "from Arctic marine biogenic sources"
6) p. 2, line 10, hyphenate "tracer-based"
7) p. 2., line 15, change to "Rising temperatures, increasing twice as fast"
8) p. 2, line 19, change to "several studies base on either in situ or"
9) p. 3, line 4, change "supposed" to "expected"
10) p. 3, line 5, change "Unusual" to "Unusually"
11) p. 3, line 10, remove the comma.

11) p. 3. Explain the relationship between the Arctic front and the polar dome. Is the polar dome the entire dome-shaped structure, while the Arctic front is the surface manifestation of the polar dome? Is one strictly based on thermal structure while the other is a barrier to transport? Are they sometimes the same and sometimes not? This ambiguity should be directly addressed in this section. It also should be clarified in the discussion at the end.

In our understanding the Arctic Front is the boundary of the polar dome. While the polar dome is considered as the three dimensional cold pool over the Arctic, the Arctic front is rather the transition region between the cold and even colder air masses. The terminology of the Arctic front is rather similar to the polar front but for different air masses. It is further noted that such a front has its strongest signature often close to the surface and is not necessarily equally established at each longitude. Moreover, there is no common definition of the term Arctic front in the literature.

We highlight our understanding of the connection between polar dome and Arctic front as follows in a revised manuscript:

*p4, l1 (see also comment 13 below): "….(Barrie, 1986; Klonecki, 2003; Stohl, 2006). In this study we regard the polar dome as the three dimensional volume of cold air above the Arctic and the Arctic front as its boundary. For instance, if temperature is regarded on a constant pressure level, the Arctic front would separate cold from even colder air masses and would be located in a region of an increased horizontal temperature gradient. This is then similar to the polar front separating subtropical from mid-latitude air masses.*

12) p. 3, line 31. Fast uplift can also be generated by synoptic-scale processes such as warm conveyor belts, not just by convection
13) p. 4, line 1. A meteorologist would probably put the words "Arctic front" here instead of "polar dome". Again, this tension should be explicitly addressed
We changed the beginning of this paragraph as follows consistently with comment 11 above:
*"The high Arctic lower troposphere is in general quite well isolated from the rest of the Arctic due to the very cold air masses located in this region. This region is referred to as the polar dome and is characterized by strong temperature contrasts at or near the surface (Arctic front) and sloping isentropes Θ, as a result of radiative cooling in the high Arctic, especially during the winter months without sunlight (Barrie, 1986; Klonecki, 2003; Stohl, 2006)."*

14). p. 4, line 8. ". . . as those air masses are cold enough to enter the high Arctic lower troposphere." Perhaps a better description is that the polar dome (Arctic air mass) wobbles over high-midlatitude sources, allowing pollution to be emitted directly into the polar dome. Do you agree?
We changed this sentence as follows:
*"As a consequence, the "Arctic haze" phenomenon is mainly fed by northern Eurasian pollution sources, when the Arctic front moves southwards since these air masses can drop to temperatures as cold as those in the high Arctic lower troposphere and thus can become part of the polar dome (Carlson, 1981; Rahn, 1981; Raatz, 1985; Iversen, 1984; Barrie, 1986; Brock et al., 1990; Dreiling and Friederich, 1997)."*

15) p. 4, line 20, remove "convective"
16) p. 4, lines 30-33. Again, ambiguity in the relationship between the polar dome boundary and the Arctic front.
We hope that clarifying the terminology of polar dome/Arctic front above helps to resolve the ambiguity here. Moreover, here in the manuscript we refer to the study of Klonecki et al. (2003) and Stohl (2006) who define the Arctic front/polar dome slightly differently than in our manuscript.

17) p. 5, line 3, hyphenate "tracer-based"
18) p. 5, line 4, remove "(Section 2,3,4)"
19) p. 5, line 11, remove "climate"
20) p. 5, line 11, hyphenate "aircraft-based"
21) p. 5, line 19, define "polynya"
22) p. 6, line 2, define "PAMARCMiP"
23) p. 6, line 19, remove "In the rear section of the instrument," (not necessary)
24) p. 7, lines 1-2, remove the sentence beginning, "From the AIMMS-20 data set, . . . ." (not needed)

25) p. 7, lines 11-14, remove sentences beginning "We calculated. . ." and "These instrumental drifts . . . ." and just state the total uncertainty in the following sentence.

We would prefer to keep the more detailed explanation for the error analysis, since relative changes are essential for our study. These can be resolved with very high precision even if the total uncertainty is large. Only stating the total uncertainty without giving the calculation specification could lead to misinterpretation of the numbers.

26) p. 7, line 14, remove "Hence"
27) p. 7, line 18, hyphenate "temperature-controlled"
28) p. 7, line 20, remove sentence beginning "The instrument itself was mounted . . . " (unnecessary detail)
29) p. 7, line 24, remove the sentence beginning "Using the same methodology as for CO. . . "

See reply to comment 25.

30) p. 8, line 2, is this the bulk or gradient Richardson number?

This is the gradient Richardson number. The manuscript was changed accordingly.

31) p. 10, line 6, is this really transport into the Arctic lower troposphere, or movement of the polar dome over emission sources?

It is most probably both. During the winter season the location of the polar dome moves further south over potential emission regions. Towards spring more frequent transport occurs since the boundary of the polar dome becomes weaker.

32) Fig. 4 caption, what sort of running means are the lines (calculated over how many points?)

It is a 10 point running mean. The caption was changed accordingly.

33) p. 11, line 2, CO increases not only because the sink (OH) decreases, but because there are continuing sources in the Arctic air mass as the front wobbles over emission regions.

We fully agree (see reply to comment 31) and we changed the manuscript as follows:
*Hence, CO increases over the course of the winter, in particular within the polar dome that furthermore expands over potential emission regions in more southern latitudes (Novelli et al., 1998; Engvall et al., 2008).*

34) p. 11, line 6, not only are there seasonally smaller emission of CO in midlatitude sources, but there is less emission directly in to the polar dome because the Arctic air mass does not wobble over these sources (it's too far poleward)

The sentence was modified as follows:
*During the transition from spring to summer (April to June) photochemical activity in the Arctic and smaller mid-latitude emissions of CO directly into the polar dome lead to decreasing CO in the Arctic until a minimum is reached at the end of the summer (Barrie et al., 1986; Klonecki et al., 2003; Engvall et al., 2008).*

35) p. 12, line 3, define "low-valued" and change "isentrops" to "isentropes". Many isentropes reach the surface; how do you determine which ones define the polar dome boundary? Is it where there is a steep surface gradient in isentropes (i.e., the Arctic front)? Again, this is related to the ambiguity in defining the polar dome vs. the Arctic front that needs to be addressed.

We removed ''low-valued''. We do not determine the location of the polar dome here, since this will be done later in the manuscript. Here we refer to the concept of the polar dome pointing out

that there is a dome like structure evident from the measured potential temperature during the measurement campaigns.

36) p. 14, line 4, hyphenate "tracer-based"
37) p. 15, line 4, remove "now"
38) p. 15, line 5, remove "Hence,"
39) p. 17, line 10, hyphenate "tracer-derived"
40), p. 17, line 14, hyphenate "tracer-derived"
41) p. 18, line 4, insert comma after "Willis et al., (2019)"
42) p. 18, line 4, hyphenate "altitude-dependent"
43) p. 20, line 5, hyphenate "low-level"
44) p. 21, Fig. 12. The trajectories in Fig. 12c are plotted against which axis, the left or the right? Make clear in the figure caption. The caption describes panels c and d as a "vertical cross section" but there is no altitude axis in either; they are time-pressure cross sections, I guess.
Figures (c) and (d) show height-time cross-sections with pressure representing the altitude of the trajectories. We changed the caption accordingly

45) p. 23, line 17, change "week" to "weak"
46) p. 26, figure caption. In the last sentence of the caption, I don't see any obvious "enhancement in BC" within the polar dome in panel d. It just looks relatively uniformly speckled, and tending toward the blue colors (lower BC values)
We changed the caption as follows:
*There is a slight enhancement in BC south of 80°N, which could be associated to local pollution.*

47) p. 27, lines 15-18. Here you seem to distinguish between the polar dome boundary and the Arctic front. You say "in comparison to the location of the Arctic front, our analysis. . . ." What position of the Arctic front are you talking about? A climatological position? A position for the NETCARE time period? Wouldn't one expect movement of either/both with synoptic-scale disturbances?
The sentence was changed as follows:
*In comparison to the location of the Arctic front determined in Klonecki et al. (2003), our analysis seems to give a more northern boundary for both July 2014 and April 2015.*

48) p. 28, line 2, hyphenate "transport-controlled"
49) p. 28, line 21, change to "is clearly separate"
50) p. 28, line 30, change to "can be identified based on "
51) p. 29, line 3, hyphenate "tracer-derived"
52) p. 29, line 5, change to "is separate from"

[revised manuscript text omitted]